# Assimilating synthetic Biogeochemical-Argo and ocean colour observations into a global ocean model to inform observing system design

David Ford[1]

[1]Met Office, FitzRoy Road, Exeter, EX1 3PB, UK

**Correspondence:** David Ford (david.ford@metoffice.gov.uk)

**Abstract.** A set of observing system simulation experiments was performed. This assessed the impact on global ocean biogeochemical reanalyses of assimilating chlorophyll from remotely sensed ocean colour, and in situ observations of chlorophyll, nitrate, oxygen, and pH from a proposed array of Biogeochemical-Argo (BGC-Argo) floats. Two potential BGC-Argo array distributions were tested: one where biogeochemical sensors are placed on all current Argo floats, and one where biogeochemical sensors are placed on a quarter of current Argo floats. Assimilating BGC-Argo data greatly improved model results throughout the water column. This included surface partial pressure of carbon dioxide ($pCO_2$), which is an important output of reanalyses. In terms of surface chlorophyll, assimilating ocean colour effectively constrained the model, with BGC-Argo providing no added benefit at the global scale. The vertical distribution of chlorophyll was improved by assimilating BGC-Argo data. Both BGC-Argo array distributions gave benefits, with greater improvements seen with more observations. From the point of view of ocean reanalysis, it is recommended to proceed with development of BGC-Argo as a priority. The proposed array of 1000 floats will lead to clear improvements in reanalyses, with a larger array likely to bring further benefits. The ocean colour satellite observing system should also be maintained, as ocean colour and BGC-Argo will provide complementary benefits.

## 1 Introduction

Throughout the ocean, physical and chemical processes interact with a teeming ecosystem to affect all life on Earth. The upwelling of nutrient-rich waters fuels the growth of phytoplankton, which form the base of the marine food web and contribute half the planet's primary production (Field et al., 1998). Oxygen is required for marine and terrestrial life, and its availability depends on ocean circulation, solubility, and biological activity. Carbon is taken up at the sea surface, at a rate contingent on physics and biology, and transported throughout the ocean. Some is stored for centuries at vast depths, mitigating climate

change. Some is quickly released back to the atmosphere. All these phenomena are regulated by an array of processes which display variability on a range of scales from milliseconds to millennia, and from nanometres to ocean basins.

Understanding, monitoring, and predicting these processes is key to addressing some of the biggest challenges facing humanity. Rising carbon dioxide ($CO_2$) emissions are leading to climatic changes which threaten severe impacts on people and ecosystems (IPCC, 2014). Uptake of carbon by the global ocean acts to mitigate these impacts, but the ocean carbon sink is highly variable and its future magnitude uncertain (McKinley et al., 2017). At the same time, when $CO_2$ dissolves in seawater it reacts with it, leading to a decrease in pH referred to as ocean acidification (Doney et al., 2009). This could have major

consequences for marine life, particularly organisms which form calcium carbonate shells, which become at risk of dissolution if the seawater pH is too low. Changes in climate and eutrophication also appear to be leading to expanding "dead zones" in the ocean (Diaz and Rosenberg, 2008; Altieri and Gedan, 2015), where oxygen concentrations are too low for most organisms to survive. On shorter time-scales, primary production varies considerably due to natural climate variability such as the El Niño Southern Oscillation (ENSO), and changes can have profound impacts on higher trophic levels, and hence the fisheries and

aquaculture on which an estimated 12 % of the global population rely for their livelihoods (FAO, 2016). All these factors and more are captured in a drive towards "Good Environmental Status" of national waters, as regulated by policies such as the Marine Strategy Framework Directive (MSFD) of the European Union (EU).

Comprehensively monitoring all relevant processes in the global ocean, and their variability and trends, is not a trivial task. For ocean biogeochemistry, the global observing system consists of various components which, while often sparse and

disparate, have allowed fundamental insights. Two decades of routine satellite ocean colour data (Groom et al., 2019) have yielded unprecedented knowledge about phytoplankton variability (Racault et al., 2017), and even helped overturn decades of scientific consensus on bloom formation (Behrenfeld and Boss, 2014). In situ stations such as the Bermuda Atlantic Time Series (BATS) have allowed long-term monitoring of multiple variables at fixed locations (Bates et al., 2014), and various networks of ships, gliders, and moorings give ongoing views of different aspects of the global ocean (Telszewski et al., 2018).

These observation networks are vital, and have transformed our understanding of ocean biogeochemistry. But they remain sparse, and coverage is insufficient to address all outstanding scientific questions, or provide comprehensive monitoring on a global scale.

Observation of ocean physics has been revolutionised by the advent of Argo (Roemmich et al., 2019). A global array of around 4000 autonomous floats drift at a typical parking depth of 1000 m, and every ten days descend to 2000 m before rising

to the surface, profiling temperature and salinity as they do so. The data are then transmitted to satellites in near-real-time, before the float returns to its parking depth. Argo has facilitated breakthroughs in climate science (Wijffels et al., 2016), and improvements in physical ocean reanalyses and forecasts (Davidson et al., 2019).

The Argo initiative is now being extended to biogeochemistry through the Biogeochemical-Argo (hereafter BGC-Argo) programme (Biogeochemical-Argo Planning Group, 2016; Roemmich et al., 2019). In the next decade, it is planned to establish a

global array of 1000 BGC-Argo floats, which are Argo floats equipped with biogeochemical sensors. The aim is for all these floats to measure six core variables: oxygen concentration ($O_2$), nitrate concentration ($NO_3$), pH, chlorophyll-a concentration (Chl-*a*), suspended particles, and downwelling irradiance. This promises to transform scientific understanding of ocean bio-

geochemistry. Thanks to a series of regional programmes, there are already over 300 operational floats measuring one or more biogeochemical variables. Few of these floats yet measure all the core BGC-Argo variables, and spatial coverage is highly un-
even, but important scientific discoveries regarding phytoplankton, carbon, and nutrient dynamics have been made (Roemmich et al., 2019).

The value of observations can be further enhanced by combining them with numerical models using data assimilation (Kalnay, 2003). Ocean colour data are increasingly assimilated in state-of-the-art reanalysis (Rousseaux and Gregg, 2015; Ciavatta et al., 2016; Ford and Barciela, 2017) and forecasting (Teruzzi et al., 2014; Skákala et al., 2018) systems. This has
consistently been shown to improve simulations of phytoplankton, but the impact on other model variables, especially sub-surface, is limited (Gehlen et al., 2015). Physical data assimilation has the potential to improve biogeochemistry, but has often been found to have the opposite effect, due to spurious impacts on vertical mixing to which biogeochemical variables are particularly sensitive (Park et al., 2018; Raghukumar et al., 2015). Assimilating multivariate in situ biogeochemical data should help address these issues and greatly improve reanalyses and forecasts (Yu et al., 2018), but due to the sparsity of observational
coverage, efforts have largely been limited to parameter estimation (Schartau et al., 2017), 1D models (Torres et al., 2006), individual research cruises (Anderson et al., 2000), or surface-only carbon data (Valsala and Maksyutov, 2010; While et al., 2012). Furthermore, in situ biogeochemical observations are rarely available in near-real-time, limiting their suitability for operational applications.

The increasing availability of BGC-Argo data promises to change this, with great potential for improving reanalyses and
forecasts (Fennel et al., 2019). For instance, BGC-Argo observations of $O_2$ have been assimilated by Verdy and Mazloff (2017), who produced a five-year state estimate of the Southern Ocean using an adjoint method, and were able to capture over 60% of the variance in oxygen profiles at 200 m and 1000 m depth. Furthermore, Cossarini et al. (2019) assimilated BGC-Argo profiles of Chl-*a* into a model of the Mediterranean Sea, and found this was successful in adjusting the shape of chlorophyll profiles, and that with the present number of BGC-Argo floats they could constrain phytoplankton dynamics in up to 10% of
the Mediterranean Sea.

This paper describes the development of a scheme to assimilate profiles of Chl-*a*, $NO_3$, $O_2$, and pH into an updated version of the Met Office's global physical–biogeochemical ocean reanalysis system. A set of observing system simulation experiments (OSSEs) (Masutani et al., 2010) is presented to assess the potential value of different numbers of BGC-Argo floats. The work forms part of a coordinated effort within the EU Horizon 2020 research project AtlantOS (https://www.atlantos-h2020.eu).
Four groups performed OSSEs assessing physics observations, the results of which have been synthesised by Gasparin et al. (2019). Two groups performed OSSEs assessing biogeochemistry, Germineaud et al. (2019) and this study. Germineaud et al. (2019) presented a probabilistic evaluation at a single assimilation time step, finding that Chl-*a* from BGC-Argo floats added value at surface locations where ocean colour was unavailable, and at depth.

The biogeochemistry OSSEs consider two potential scenarios: 1) a global BGC-Argo array equivalent to having biogeo-
chemical sensors on one in four existing Argo floats, which is comparable to the planned target of 1000 floats, and 2) a global BGC-Argo array equivalent to having biogeochemical sensors on all existing Argo floats. The aims were to assess the impact on reanalysis and forecasting systems that might be seen by assimilating multivariate BGC-Argo data, the influence of array

size, and the value BGC-Argo would add to the existing ocean colour satellite constellation. Assimilation of physics variables was not included, due to the issues mentioned above, and reflecting the way state-of-the-art biogeochemical reanalyses are run
(Fennel et al., 2019).

This paper describes the updated model and newly-developed assimilation scheme, and setup of the OSSEs. Results are then presented showing the impact of assimilating the two potential BGC-Argo arrays, with and without ocean colour data. Finally, recommendations are made for the future development of observing and assimilation systems.

## 2 Model and assimilation

The reanalysis system is an upgraded version of that used in previous biogeochemical data assimilation studies at the Met Office (Ford et al., 2012; While et al., 2012; Ford and Barciela, 2017; Ford, 2020).

### 2.1 Model

The physical ocean model used is the GO6 configuration (Storkey et al., 2018) of the Nucleus for European Modelling of the Ocean (NEMO) hydrodynamic model (Madec, 2008), using the extended ORCA025 tripolar grid, which has a horizontal
resolution of 1/4° and 75 vertical levels. This is coupled online to the GSI8.1 configuration (Ridley et al., 2018) of the Los Alamos Sea Ice Model (CICE) (Hunke et al., 2015). Together, these form the ocean and sea ice components of the GC3.1 configuration (Williams et al., 2017) of the Hadley Centre Global Environment Model version 3 (HadGEM3), which is used for physical climate simulations submitted to the Coupled Model Intercomparison Project Phase 6 (CMIP6) (Eyring et al., 2016). When combined with the physics version of the data assimilation scheme described below, the ocean and sea ice models
are also used in version 14 of the Forecasting Ocean Assimilation Model (FOAM), earlier versions of which are described by Blockley et al. (2014) and Storkey et al. (2010). FOAM is run operationally at the Met Office to produce short-range forecasts of the physical ocean and sea ice state. It is also used to initialise the ocean and sea ice components of the Met Office Global Seasonal forecasting system version 5 (GloSea5) (MacLachlan et al., 2015; Scaife et al., 2014), and short-range coupled ocean–atmosphere forecasting system (Guiavarc'h et al., 2019).

The biogeochemical ocean model used in this study is version 2 of the Model of Ecosystem Dynamics, nutrient Utilisation, Sequestration and Acidification (MEDUSA) (Yool et al., 2013). MEDUSA is of intermediate complexity, representing two phytoplankton and two zooplankton types, and the cycles of nitrogen, silicon, iron, carbon, and oxygen. This differs from previous versions of the Met Office physical–biogeochemical ocean reanalysis system (Ford and Barciela, 2017), which used the Hadley Centre Ocean Carbon Cycle Model (HadOCC) (Palmer and Totterdell, 2001). This is because, following an in-
tercomparison (Kwiatkowski et al., 2014) of biogeochemical models developed in the UK, MEDUSA was chosen to be the ocean biogeochemical component of version 1 of the UK Earth System Model (UKESM1) (Sellar et al., 2019). UKESM1 consists of a lower-resolution version of GC3.1, coupled with models of ocean biogeochemistry, atmospheric chemistry and aerosols, and ice-sheets, and is used for Earth system climate simulations submitted to CMIP6. Using the same model versions

for forecasting, reanalysis, and climate simulations provides a seamless framework for simulating the Earth system (Martin
et al., 2010).

## 2.2 Assimilation

### 2.2.1 Overview

The data assimilation scheme used here is version 5 of NEMOVAR (Weaver et al., 2003, 2005; Mogensen et al., 2009, 2012),
following the implementation for assimilating physical variables into the global FOAM system (Waters et al., 2015), and
for assimilating ocean colour data into HadOCC (Ford, 2020) and the European Regional Seas Ecosystem Model (ERSEM)
(Skákala et al., 2018, 2020). As detailed in Waters et al. (2015), this implementation of NEMOVAR uses a first guess at
appropriate time (FGAT) 3D-Var methodology. A conjugate gradient algorithm is used to iteratively minimise the cost function

$$J(\delta\mathbf{x}) = \frac{1}{2}\delta\mathbf{x}^{\mathrm{T}}\mathbf{B}^{-1}\delta\mathbf{x} + \frac{1}{2}(\mathbf{d} - \mathbf{H}\delta\mathbf{x})^{\mathrm{T}}\mathbf{R}^{-1}(\mathbf{d} - \mathbf{H}\delta\mathbf{x}) \tag{1}$$

where the increment $\delta\mathbf{x} = \mathbf{x} - \mathbf{x}_{\mathrm{b}}$ is the difference between the state vector $\mathbf{x}$ and its background estimate $\mathbf{x}_{\mathrm{b}}$, the innovation
vector $\mathbf{d} = \mathbf{y} - H(\mathbf{x}_i)$ is the difference between the observation vector $\mathbf{y}$ and $\mathbf{x}_i = \mathbf{M}_{\mathrm{t}_0 \to \mathrm{t}_i}(\mathbf{x}_{\mathrm{b}})$, where $\mathbf{M}_{\mathrm{t}_0 \to \mathrm{t}_i}$ is the nonlinear
propagation model that propagates the background state to the state at time $i$, operated on by the observation operator $H$, $\mathbf{H}$
is the linearised observation operator, $\mathbf{B}$ is the background error covariance matrix, and $\mathbf{R}$ is the observation error covariance
matrix. A diffusion operator is used to efficiently model spatial correlations (Mirouze and Weaver, 2010; Mirouze et al., 2016).
Ten iterations of the diffusion operator are applied, simulating the matrix multiplication of an autoregressive correlation matrix,
which provides a good approximation to a Gaussian correlation function (Waters et al., 2015). The observation operator forms
part of the NEMO code, and computes model values in observation space by interpolating model fields to observation locations
at the closest model time step to the time each observation was made. The observation operator was extended in this study to
work for 3D biogeochemical variables as well as physical variables.

When applied to physics data, NEMOVAR decomposes the full multivariate background error covariance matrix into an
unbalanced and a balanced component for each variable. The unbalanced component considers the uncorrelated component of
each variable using univariate error covariances, while the balanced component considers correlations between variables. The
balanced component is derived using a set of linearised balance operators, based on physical relationships (Weaver et al., 2005;
Mogensen et al., 2012). In this study, NEMOVAR has been applied to biogeochemical variables with no multivariate relation-
ships applied, and the cost function is minimised separately for each assimilated variable. Development of biogeochemical
balance relationships within NEMOVAR could be expected to bring improved results, but this would be a major development
to NEMOVAR. The aim of this study was to develop an initial implementation that could be used with BGC-Argo data, and
that can be further developed as systems mature.

All increments are applied to the model over one day using incremental analysis updates (IAU) (Bloom et al., 1996), which
applies an equal proportion of the increments at each model time step, in order to reduce initialisation shocks.

NEMOVAR is used in this study to assimilate simulated ocean colour and BGC-Argo data, as described in the following sections. NEMOVAR can be used for combined physical–biogeochemical assimilation (Ford, 2020), but physics data was not assimilated in this study.

### 2.2.2    Ocean colour

NEMOVAR was used here to assimilate total surface $\log_{10}$(Chl-*a*) from ocean colour. Since MEDUSA simulates Chl-*a* for two
phytoplankton types, diatoms and non-diatoms, these were summed by the observation operator to give total Chl-*a*, to match the input observations. Log-transformation was performed in order to give a more Gaussian error distribution (Campbell, 1995). The background and observation error covariances used were the same as in Ford (2020). In the horizontal, a correlation length-scale based on the first baroclinic Rossby radius was used, varying from a value of 25 km at low latitudes to 150 km at the Equator, consistent with Waters et al. (2015).

For surface data, such as ocean colour, NEMOVAR can be applied in one of two ways. The first, which is computationally most efficient and has been used in previous ocean colour assimilation studies (Ford, 2020; Skákala et al., 2018, 2020), is to calculate a set of 2D surface increments which are applied equally through the mixed layer. The second is to calculate a set of 3D increments, with information from the surface observations propagated downwards using vertical correlation length-scales, as described by Waters et al. (2015) for physical variables. The sub-surface background error standard deviations are
parameterised based on the vertical gradient of density with depth to allow a flow-dependent vertical structure to the errors. The vertical correlation length-scale is dependent on the model's mixed layer depth, as determined from a one-day model forecast. At the surface, the vertical correlation length-scale is set to the depth of the mixed layer, so that information from surface observations is spread to the base of the mixed layer but not below it. The latter method allows satellite and in situ profile observations of a given variable to be consistently combined by NEMOVAR, to produce a single set of 3D increments
for that variable, and was therefore the method employed in this study.

    This gives a set of 3D $\log_{10}$(Chl-*a*) increments on the model grid, which must be applied to the model. The $\log_{10}$(Chl-*a*) increments were converted to Chl-*a* increments using the background total Chl-*a*, and split between diatoms and non-diatoms so as to conserve the ratio between the two in the background model field. Phytoplankton biomass was then similarly updated, to conserve the stoichiometric ratios in the background field, following the approach of Teruzzi et al. (2014) and Skákala et al.
(2018, 2020).

### 2.2.3    BGC-Argo

For in situ profiles of biogeochemistry, as might be obtained from BGC-Argo, sets of 3D increments were calculated for each assimilated variable, following the physics implementation of Waters et al. (2015). The method was the same as for calculating 3D ocean colour increments, as described above. The vertical correlation length-scale was flow-dependent and varies with
depth, as detailed by Waters et al. (2015). At the surface the vertical correlation length-scale was set to the depth of the mixed layer, decreasing to a minimum value at the base of the mixed layer. This minimised the spread of information across the pycnocline, due to the lack of correlation of water mass properties in and below the mixed layer (Waters et al., 2015; Fontana

et al., 2013). Below the mixed layer, the vertical correlation length-scale increased with depth, proportional to the increase in vertical model grid spacing that occurs with depth.

In this study Chl-$a$, $NO_3$, $O_2$, and pH were assimilated, but the methodology is simple to extend to other model variables. As for ocean colour assimilation, Chl-$a$ is the sum of diatom and non-diatom Chl-$a$, and a log-transformation was performed prior to assimilation. As described above, assimilation of Chl-$a$ from ocean colour and in situ profiles can be combined. $NO_3$ and $O_2$ are state variables in MEDUSA, taking $NO_3$ to be equivalent to the MEDUSA DIN variable, while pH is a diagnostic variable calculated using version 2.0 of the mocsy carbonate package (Orr and Epitalon, 2015), which implements the SolveSAPHE algorithm (Munhoven, 2013) for solving the alkalinity-pH equation.

The Chl-$a$ increments were applied using the stoichiometric balancing method described for ocean colour above. The $NO_3$ increments were directly applied to the MEDUSA DIN variable, and the $O_2$ increments to the $O_2$ variable. As pH is a diagnostic variable, the pH increments cannot be applied directly. The approach taken to the assimilation of partial pressure of $CO_2$ ($pCO_2$) into HadOCC (While et al., 2012) was therefore adopted here with pH. In HadOCC, $pCO_2$ is a function of temperature, salinity, DIC, and alkalinity, and at constant temperature and salinity constant lines of $pCO_2$ are found in DIC/alkalinity space (see Fig. 1 of While et al. (2012)). The scheme of While et al. (2012) assumes that temperature and salinity are error-free (and can be directly updated by physical data assimilation if not), and therefore updates DIC and alkalinity. As there is no unique combination of DIC and alkalinity that gives a specific $pCO_2$ value, the smallest combined change to DIC and alkalinity is made in order to reach the target $pCO_2$ value. The same approach was taken here with pH, which in MEDUSA is a function of temperature, salinity, nutrients, latitude, depth, DIC, and alkalinity. In DIC/alkalinity space, locally constant lines of pH are found when considering the range of present oceanic conditions (see Fig. 1a of Munhoven (2013)). The scheme developed here therefore updates DIC and alkalinity, assuming the other contributors to pH to be error-free, by making the smallest combined change which would give the target pH.

## 3   Observing system simulation experiments (OSSEs)

### 3.1   Overview

As detailed by Masutani et al. (2010), OSSEs provide a way to test the impact on forecasts and reanalyses of assimilating observations which do not yet exist, by using synthetic observations. An OSSE typically comprises the following elements:

- A "nature run", which is a realistic non-assimilative model simulation of the real world, which provides a "truth" against which to validate the assimilative model.

- Synthetic observations representing both existing routine observations and future observing networks, which are sampled from the nature run with appropriate errors prescribed.

- Optionally, a free run, which provides an alternative model simulation of the nature run period.

- An assimilative run, which assimilates synthetic observations representing existing routine observations into the alternative model simulation.

- One or more additional versions of the assimilative run which also assimilate synthetic observations representing the future observing networks under consideration.

- Assessment of the impact on reanalysis or forecast skill of assimilating these observations, by validating against the nature run.

One of the keys to obtaining informative results from an OSSE is to ensure that all sources of error are appropriately accounted for (Halliwell et al., 2014, 2017; Hoffman and Atlas, 2016). If the free run is more similar to the nature run than the real forecasting system is to the real world, then it can become easier for the assimilative system to recover the truth, and the impact of the observing networks may be incorrectly estimated. As such, three general OSSE approaches have been developed, which differ in how the free run varies from the nature run.

- In "identical twin experiments", the nature and free runs differ only in their initial conditions. This set-up was frequently used in early OSSEs, but as most sources of model error are neglected, the results were found to be overly optimistic, and the approach is no longer widely recommended (Arnold and Dey, 1986).

- In "fraternal twin experiments", the same model is still used for both the nature and free run, but more aspects are modified. These could include the initial conditions, lateral and surface boundary conditions, parameterisations, and resolution. This takes much better account of model errors, and the approach is recommended over identical twin experiments (Arnold and Dey, 1986; Masutani et al., 2010; Yu et al., 2019).

- In "full OSSEs", significantly different models are used for the nature and free runs, in order to make the two more independent. The nature run is often run either at higher resolution than the assimilative model, or with significantly different parameterisations (Fujii et al., 2019). It is recommended to use this approach if possible (Masutani et al., 2010), but it relies on having two appropriately different models available, which is not always the case.

Due to the lack of availability of an appropriate alternative model for the nature run, it was decided within AtlantOS to take a fraternal twin approach for the biogeochemical OSSEs. This is sufficient to account for most sources of error, as long as any limitations of the approach are considered when drawing and acting upon conclusions.

## 3.2 Model formulation

The nature run in this study was run from 1 January 2008 to 31 December 2009, using the default parameterisations for the model versions used. This is intended to be the best available non-assimilative model representation of the real world. Validation of the general performance of the different system components can be found in the references given in Section 2, and validation of the nature run is presented in Section 4.1. Atmospheric boundary conditions were taken from the ERA-Interim reanalysis

(Dee et al., 2011). Initial conditions for NEMO were taken from the end of a 30-year hindcast of GO6 (Storkey et al., 2018). Initial conditions for CICE were taken from a pre-operational trial of the FOAM v14 system. Initial conditions for MEDUSA were based on year 5000 of the initial ocean-only phase of the spin-up of UKESM1 for use in CMIP6 projections (Yool et al., 2020). As the UKESM1 spin-up was run at 1° resolution with pre-industrial atmospheric $CO_2$ concentrations, the UKESM1 fields were interpolated to 1/4° resolution, and the DIC and alkalinity fields replaced by the contemporary model estimates used to initialise the 1/4° resolution experiments in Ford (2020). To allow the model to settle, the first year of the nature run was treated as spin-up. The period was chosen to match OSSEs of the in situ physics observing system performed at the Met Office (Mao et al., 2020), and more widely as part of AtlantOS (Gasparin et al., 2019).

The free run was performed for the same period, including spin-up, but differed from the nature run in the following ways:

- Atmospheric boundary conditions were taken from the JRA-55 reanalysis (Kobayashi et al., 2015).

- NEMO initial conditions were taken from an earlier date (1 January 1999) of the hindcast of Storkey et al. (2018).

- MEDUSA initial conditions were taken from an earlier year (1218) of the UKESM1 ocean-only spin-up, with DIC and alkalinity taken from the end of the non-assimilative 1/4° resolution experiment of Ford (2020).

- The NEMO parameter rn_efr, which affects near-inertial wave breaking and therefore vertical mixing (Calvert and Siddorn, 2013), was increased from 0.05 to 0.1.

- The scheme used for advection of biogeochemical variables was changed from Total Variance Dissipation (TVD) (Zalesak, 1979) to the Monotone Upstream Scheme for Conservative Laws (MUSCL) (Van Leer, 1977; Lévy et al., 2001).

- An alternative set of MEDUSA parameters was used, specifically Parameter Set 3 from Table 2 of Hemmings et al. (2015), which was found to give differences of an appropriate magnitude.

Together, these modifications generate approximations to the errors that exist in atmospheric fluxes and simulations of ocean physics and biogeochemistry. It is important to modify all of these, as errors in atmosphere and ocean physics have significant impacts on biogeochemical reanalyses and forecasts, and these errors must be accounted for if realistic conclusions are to be drawn from the OSSEs.

### 3.3 Synthetic observations

Synthetic ocean colour and BGC-Argo observations were generated from the nature run for each day of 2009. Total Chl-*a* from ocean colour represents the current observing network typically assimilated in biogeochemical reanalyses (Fennel et al., 2019). Observations were simulated at the same locations as were actually observed in 2009 in version 3.1 of the ESA CCI level three daily merged sinusoidal grid product (Sathyendranath et al., 2018, 2019), as used in recent Met Office reanalyses (Ford and Barciela, 2017). Whilst the products date from 2009 rather than present day, the observational coverage is similar, with three sensors contributing in 2009 (MERIS, MODIS, and SeaWiFS), and three contributing to recent ocean colour products (MODIS, OLCI, and VIIRS). BGC-Argo float trajectories were based on Argo float trajectories (Argo, 2000) used for physics OSSEs

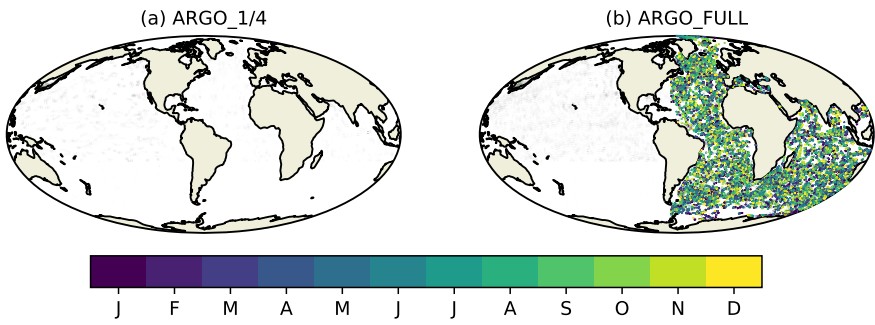

**Figure 1.** Simulated BGC-Argo float trajectories for 2009 equivalent to having biogeochemical sensors on (a) one in four Argo floats and (b) all Argo floats. Colours represent the month in which each observation is valid.

within AtlantOS (Gasparin et al., 2019). These were generated using recent real Argo float trajectories, with modifications to
280 ensure more even geographic coverage - for details see Gasparin et al. (2019). In this study, for testing the scenario equivalent to having biogeochemical sensors on all current standard Argo floats, the same "backbone" float trajectories were used as in the studies synthesised by Gasparin et al. (2019). For the scenario equivalent to having biogeochemical sensors on one in four Argo floats, these were subsampled using the last two digits of the float IDs. The geographic coverage in each case is shown in Fig. 1.

In data assimilation, two components of observation error are typically considered: measurement error and representation error (Janjić et al., 2018). Measurement error has been accounted for in this study by adding unbiased Gaussian noise to the nature run values at observation locations, using standard deviations from the literature. A standard deviation of 30 % was agreed on within AtlantOS for Chl-*a* from ocean colour, a value commonly used in assimilation studies (Pradhan et al., 2020). The same value was used for BGC-Argo Chl-*a* profiles, consistent with Boss et al. (2008). For the remaining variables the
values from Johnson et al. (2017) were used: 1 % for $O_2$, 0.005 for pH, and 0.5 mmol m$^{-3}$ for $NO_3$. To avoid generating spuriously noisy profiles, a single value of Gaussian noise was calculated per profile, rather than at every depth level. Where the standard deviations used were a percentage, this was calculated using the mean of the profile.

Representation error arises from observations and models representing differing spatial and temporal scales and processes. Since the nature and free runs were at the same resolution, this was accounted for in the same way as for the physics OSSEs
in AtlantOS (Gasparin et al., 2019). For each profile, the equivalent nature run values were calculated either three days before or three days after, chosen at random. The difference between these and the "truth" value were taken to be the representation error, and added on to the observation values. The advantage of this approach is that representation error is higher in more variable regions, as would be expected in real-world data assimilation applications.

## 3.4 Error covariances

For assimilating ocean colour data, the monthly-varying background and observation error standard deviations from Ford (2020) were used. To provide consistency between assimilating surface and in situ $\log_{10}$(Chl-*a*), these were also used for assimilating $\log_{10}$(Chl-*a*) from BGC-Argo.

For other variables, pre-existing error standard deviations were not available, so were calculated for this study. Observation error standard deviations were set to a climatological constant equal to the average global observation error specified. These were fixed in time, and specified as 0.638 mmol m$^{-3}$ for $NO_3$, 2.767 mmol m$^{-3}$ for $O_2$, and 0.006 for pH. Background error standard deviations were calculated using the Canadian Quick (CQ) method (Polavarapu et al., 2005; Jackson et al., 2008), which uses the variance of the difference between successive days of a free-running model simulation as a proxy for background error variance. Annual background error standard deviations were calculated from the output of the free run. The CQ method is known to underestimate the magnitude of the error standard deviations (Bannister, 2008), and the results in this study were considerably lower than the observation error standard deviations used. In order to give sufficient weight to the observations for the assimilation to be effective, the background error standard deviations were inflated. This was achieved by multiplying the gridded field of background error standard deviations for each variable by a constant, so that the global mean background error standard deviation matched the observation error standard deviation used for that variable. This meant that on average, equal weight was given to the background and to the observations, but the ratio of background to observation error varied spatially based on the estimates from the CQ method. Once the system is fully functioning with real BGC-Argo data available, the background error estimates can be appropriately refined, based on the errors in the real-world assimilative model, and the actual distribution of BGC-Argo observations.

## 3.5 Experiments

Using these inputs, a set of assimilation experiments were performed, in addition to the nature and free runs, as detailed in Table 1. The nature and free runs were run from 1 January 2008 to 31 December 2009, with the first year treated as spin-up. Each assimilation experiment was run from 1 January 2009 to 31 December 2009, using initial conditions from the end of the free run spin-up, and assimilating the synthetic observations into the version of the model used for the free run.

Five assimilation experiments were run. One just assimilated ocean colour. Two assimilated ocean colour in combination with the 1/4 subsampled BGC-Argo array and the full BGC-Argo array. A final two runs assimilated the 1/4 subsampled and full BGC-Argo arrays without ocean colour.

All the experiments, with unique identifiers for each, are detailed in Table 1.

**Table 1.** Experiments performed.

| Identifier | Assimilation |
|---|---|
| NATURE | None |
| FREE | None |
| OC | Ocean colour |
| ARGO_1/4_OC | 1/4 Argo + ocean colour |
| ARGO_FULL_OC | Full Argo + ocean colour |
| ARGO_1/4 | 1/4 Argo |
| ARGO_FULL | Full Argo |

## 3.6 Metrics

The main metrics used for assessment are the absolute and percentage reduction in median absolute error (MAE), defined respectively as:

$$MAE_{red\_abs} = MAE_{control} - MAE_{OSSE} \qquad (2)$$

$$MAE_{red\_\%} = \frac{MAE_{control} - MAE_{OSSE}}{MAE_{control}} \times 100 \qquad (3)$$

where $MAE_{OSSE}$ is the MAE of each OSSE compared with NATURE, and $MAE_{control}$ is the MAE of a control run compared with NATURE. When considering the impact of data assimilation versus a free run, FREE is used as the control run, and when assessing the added value of BGC-Argo over ocean colour, OC is used as the control run. A positive value of $MAE_{red\_abs}$ or $MAE_{red\_\%}$ represents a reduction in error in the OSSE compared to the control, and a negative value represents an increase in error. This is a modification of the approach taken by Gasparin et al. (2019), who used the percentage reduction in mean square error. MAE is used instead because the biogeochemical variables being considered are highly non-Gaussian, so it is more appropriate to use a metric such as MAE which is based on robust statistics. $MAE_{red\_abs}$ is used in addition to $MAE_{red\_\%}$, as this can be more informative in regions where $MAE_{control}$ is small.

Where $MAE_{red\_abs}$ or $MAE_{red\_\%}$ is presented as a spatial map, the MAE was calculated independently for each model grid cell. This was done by calculating the absolute difference between the model run and the nature run in that grid cell on each day of the given time period, and calculating the median of those values. Where $MAE_{red\_abs}$ or $MAE_{red\_\%}$ is presented as a profile, the MAE was calculated independently for each model depth level. At each depth, the absolute difference between the model run and the nature run on each day of the given time period was calculated for each grid cell in the region of interest. The median of this set of values was calculated, weighted by the area of each grid cell, to give the MAE value for that depth level.

## 4   Results

The results are presented in two sub-sections below. The first assesses the ability of NATURE to capture key ocean features, and how differences between NATURE and FREE compare to errors in real-world reanalyses. The second assesses the assimilation runs, and the potential impact of assimilating BGC-Argo and ocean colour data.

### 4.1   Errors in free-running model

As stated in Section 3, OSSEs yield the most reliable conclusions when all sources of real-world error have been appropriately accounted for (Halliwell et al., 2014). This means that the errors between FREE and NATURE should, ideally, be broadly similar to the errors between FREE and the real world. Furthermore, NATURE should be able to represent key features observed in the real ocean. As the real-world ocean is not known exactly, observation-based products must be used to perform this assessment, even though these can have large uncertainties and do not exactly represent the real world. For many biogeochemical variables, coarse climatologies are the only suitable products for a global assessment.

Figure 2 shows surface fields of temperature, Chl-$a$, $NO_3$, $O_2$, pH, and $pCO_2$, from observation-based products, NATURE, and FREE. These are plotted for the final month of the simulations, December 2009, which is when the largest cumulative impact from the data assimilation will be seen. The observation-based products used are the EN4.2.1 monthly analysis product for temperature (Good et al., 2013; Gouretski and Reseghetti, 2010), the Ocean Colour CCI v3.1 monthly product for Chl-$a$ (Sathyendranath et al., 2018, 2019), the 2018 World Ocean Atlas (WOA18) monthly climatology for $NO_3$ and $O_2$ (Garcia et al., 2018a, b), the GLODAP v2 annual climatology for pH (Key et al., 2015; Lauvset et al., 2016), and the monthly statistical analysis product of Landschützer et al. (2015a, b) for $pCO_2$. The observation-based products were bilinearly interpolated to the model grid.

For both physical and biogeochemical variables, NATURE captured the broad global distribution, with generally appropriate magnitudes. There were some discrepancies, such as an overestimation of Chl-$a$ in parts of the Pacific and Southern Oceans, and an underestimation in the Atlantic and Indian Oceans. $NO_3$ was too low in the Atlantic and Indian Oceans, and $pCO_2$ was too low in the Tropical Pacific. Overall though, NATURE matched the observation-based products sufficiently well for it to be used as the "truth" in these experiments.

FREE also broadly captured these features, as expected from state-of-the-art models (Fennel et al., 2019). Differences between FREE and NATURE were generally similar to the differences between FREE and the observation-based products, but with some exceptions. This can be seen more clearly in Fig. 3, which shows the absolute difference between FREE and the observation-based products, and between FREE and NATURE, for the fields plotted in Fig. 2.

For temperature (Fig. 3a-b), the absolute difference between FREE and NATURE was very similar in pattern to that between FREE and the EN4 analysis, but slightly lower in magnitude in some regions. This suggests that the perturbations applied to the physics (different atmospheric fluxes, initial conditions, and vertical mixing) resulted in an error contribution to the biogeochemical model similar to, but slightly smaller than, that seen in state-of-the-art modelling systems. For Chl-$a$ (Fig. 3c-d), the two sets of absolute difference were broadly similar in the Pacific and Southern Oceans, but in the Tropical Atlantic and Indian

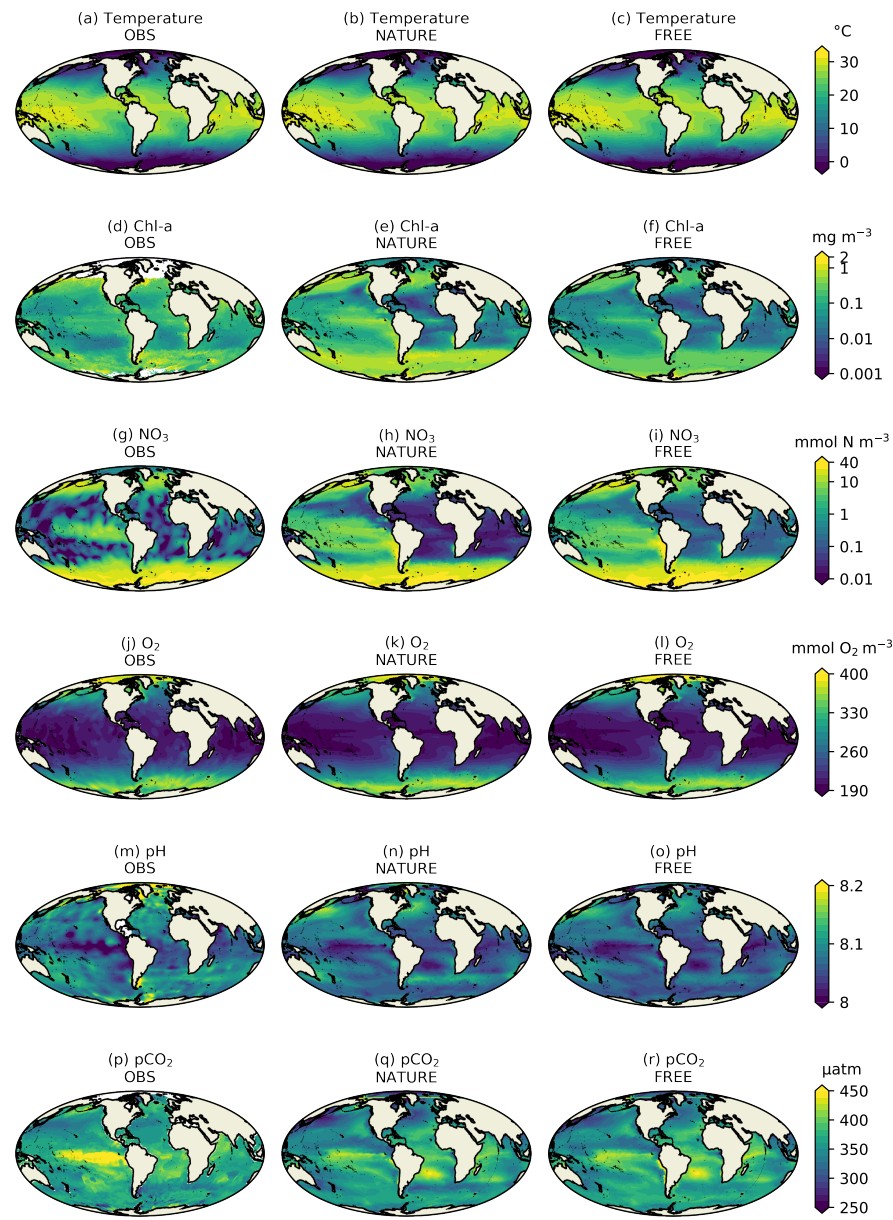

**Figure 2.** Monthly mean surface (a-c) temperature, (d-f) Chl-*a*, (g-i) $NO_3$, (j-l) $O_2$, (m-o) pH, and (p-r) $pCO_2$, for December 2009, from real-world observation-based products (left column), NATURE (middle column), and FREE (right column).

Oceans the absolute difference between FREE and NATURE was smaller than between FREE and the CCI data. In NATURE the Chl-*a* in these regions was too low compared with observations, linked to low nutrient concentrations (Fig. 2). The mod-

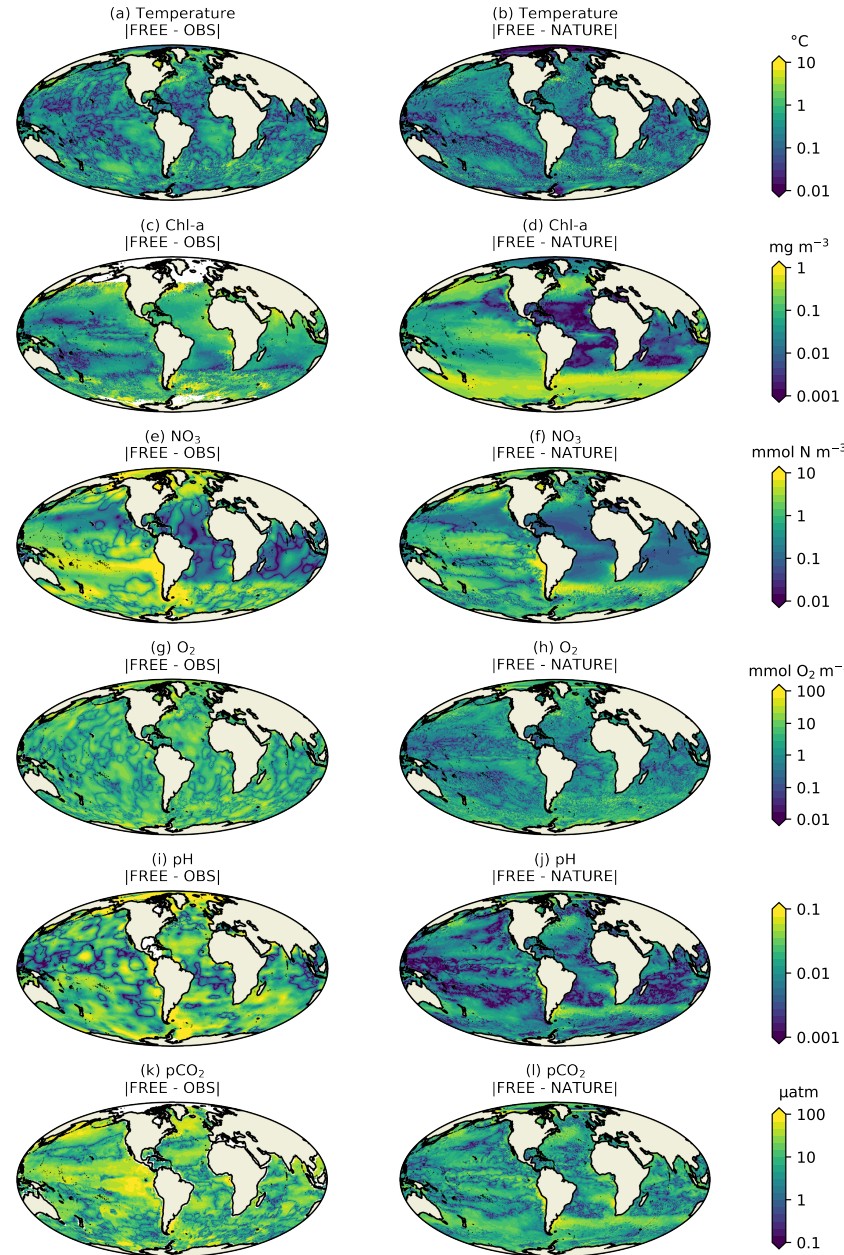

**Figure 3.** Absolute difference for December 2009 for surface (a-b) temperature, (c-d) Chl-*a*, (e-f) NO$_3$, (g-h) O$_2$, (i-j) pH, and (k-l) pCO$_2$, between FREE and real-world observation-based products (left column), and between FREE and NATURE (right column).

ifications introduced in FREE served to increase nutrient concentrations in these regions, but also to suppress phytoplankton growth, resulting in little overall change in Chl-*a*. This is largely the result of increasing the nitrogen nutrient uptake half-

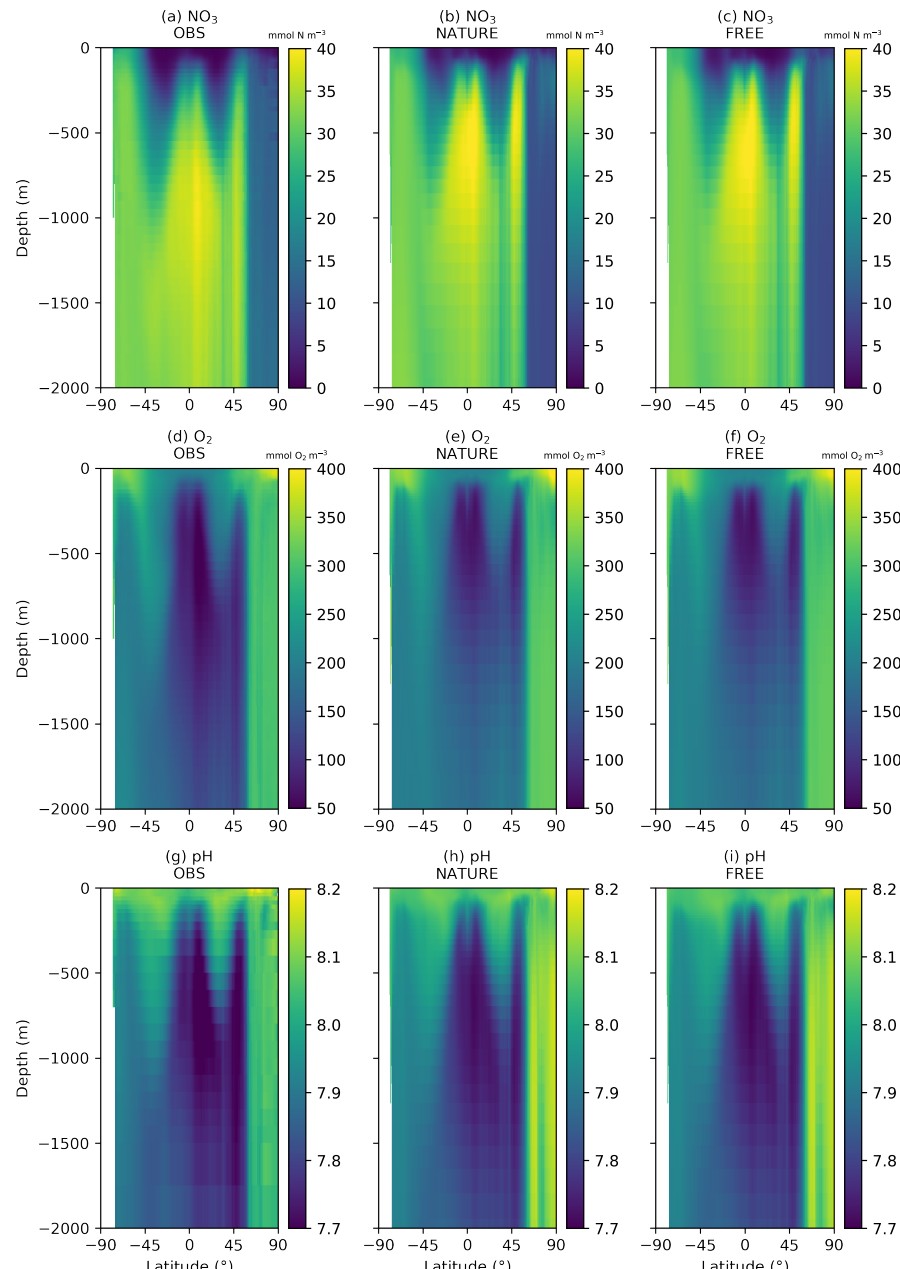

**Figure 4.** Annual zonal mean sections of (a-c) NO$_3$, (d-f) O$_2$, and (g-i) pH, from real-world observation-based products (left column), NATURE (middle column), and FREE (right column).

saturation concentration for phytoplankton, and decreasing the zooplankton grazing half-saturation concentration. Elsewhere though, the levels of absolute error were broadly similar, meaning the OSSEs had realistic levels of model error. For NO$_3$, O$_2$,

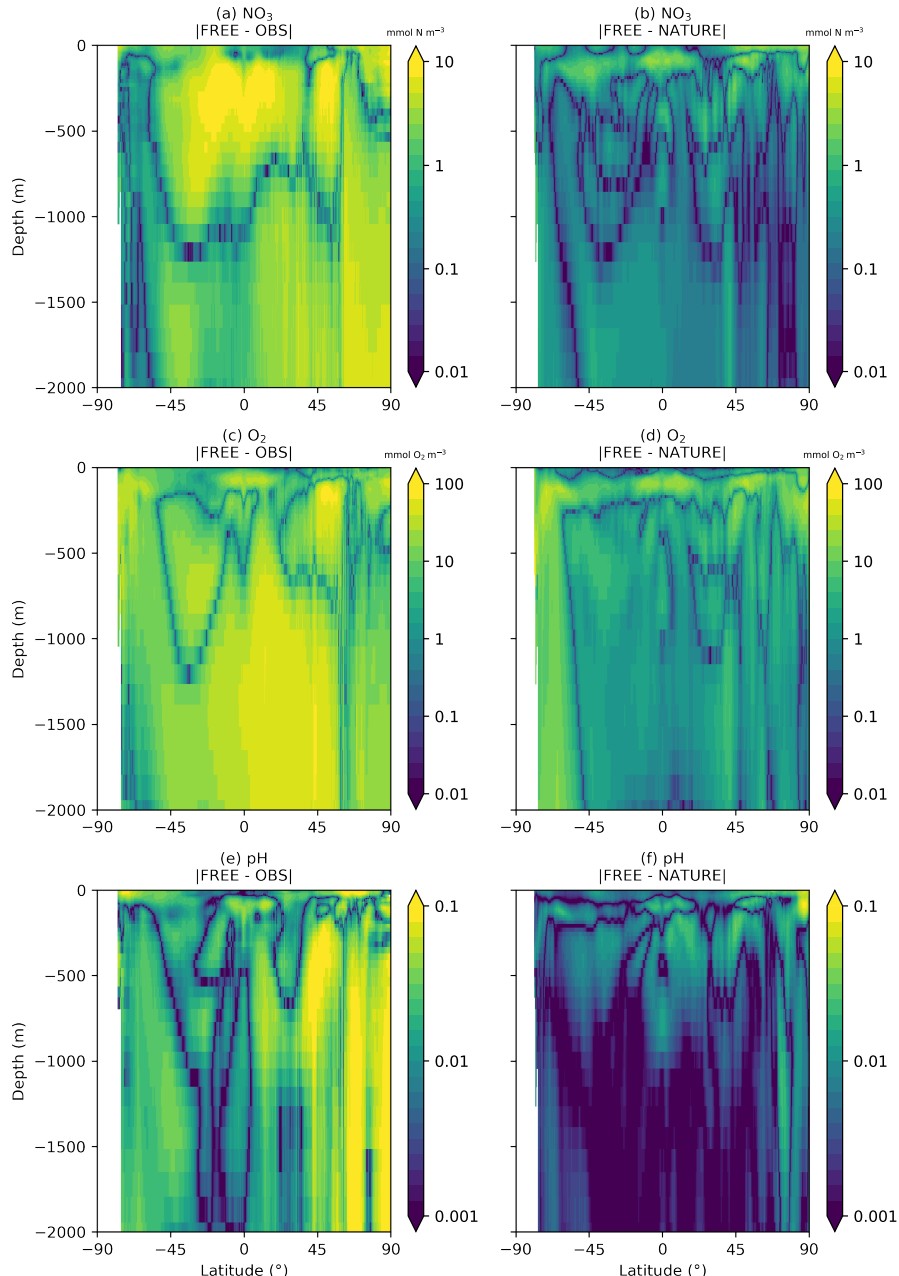

**Figure 5.** Absolute difference between FREE and real-world observation-based products (left column), and between FREE and NATURE (right column), for annual zonal mean sections of (a-b) $NO_3$, (c-d) $O_2$, and (e-f) pH.

pH, and $pCO_2$ (Fig. 3e-l), the global distributions of absolute error were generally similar between FREE and NATURE and between FREE and the observation-based products, although the absolute difference between FREE and NATURE was often

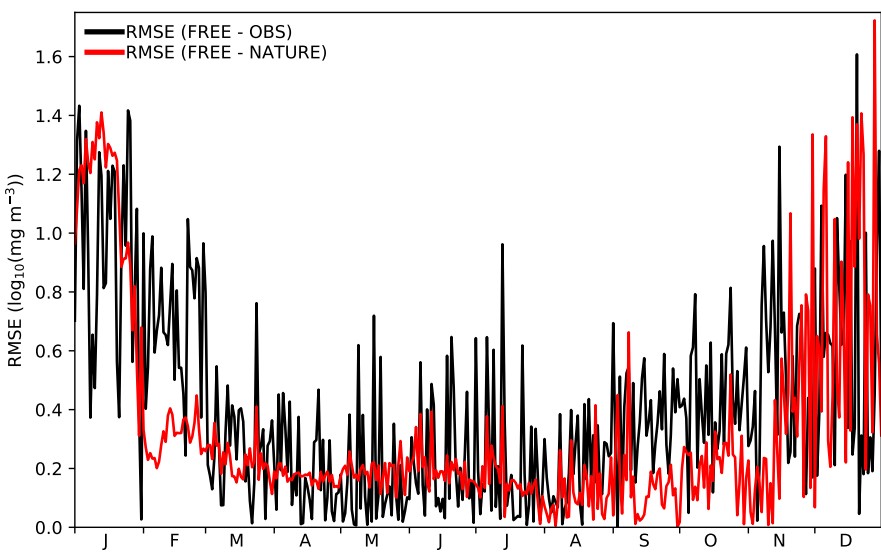

**Figure 6.** Time series of daily global RMSE for surface $\log_{10}$(Chl-$a$) between FREE and CCI ocean colour observations (black line) and between FREE and NATURE (red line) at CCI ocean colour observation locations.

smaller. It should be remembered though that the observation-based products used here have large uncertainties themselves, while NATURE is the "truth" and so error-free.

A similar comparison is required for the subsurface ocean, as BGC-Argo profiles were simulated for the upper 2000 m. Fewer observation products are available for such an assessment. No subsurface climatology is available for Chl-$a$, and only annual climatologies covering the full water column are available for $NO_3$ and $O_2$ from WOA18, and for pH from GLODAP v2. Annual zonal mean sections of $NO_3$, $O_2$, and pH are therefore plotted for the upper 2000 m in Fig. 4, from the observation-based products, NATURE, and FREE. For all three variables, NATURE captured well the broad zonal and depth variations of

the climatologies, with appropriate magnitudes. $NO_3$ concentrations in the upper 1000 m were slightly too high in the low- and mid-latitudes, as was $O_2$ and pH below 500 m, but there were no major discrepancies. FREE also captured the observed distibutions of all three variables, though differences between FREE and NATURE were often slightly smaller than between FREE and the observation-based products, as shown in Fig. 5.

    A further consideration is that the growth of errors with time between FREE and NATURE should be comparable to that

between FREE and real-world observations (Halliwell et al., 2014). The only assimilated variable for which there are sufficient daily observations to assess this is Chl-$a$ from ocean colour. In Fig. 4 time series of root mean square error (RMSE) for surface $\log_{10}$(Chl-$a$) are plotted between FREE and NATURE and between FREE and observations. The observations are the daily CCI data used to define the locations of the synthetic ocean colour observations. For the comparison, model values were bilinearly interpolated to the observation locations, and a daily global RMSE value calculated. The magnitude of the RMSE between

FREE and NATURE and between FREE and the observations was very similar, and remained comparable throughout the year,

with higher RMSE in austral summer in both cases. The RMSE variability was typically smaller between FREE and NATURE though, suggesting this to be a source of error not fully captured in FREE.

While it is not ideal that the Chl-*a* errors differ in the Tropical Atlantic and Indian Oceans, and errors between FREE and NATURE were too low for some variables, achieving globally appropriate levels of error with a complex biogeochemical model with globally uniform parameterisations could not be managed within the resources of the project. Furthermore, the similarity of Chl-*a* between NATURE and FREE in some regions was due to the introduction of compensating errors in FREE, rather than a lack of model error. This itself is a common feature of reanalyses, which can result in data assimilation increasing overall error by correctly reducing one of a set of compensating errors, as demonstrated by Ford and Barciela (2017). For regions and variables where the errors between FREE and NATURE were too low, the potential result may be to underestimate the impact of assimilating dense data, in this case ocean colour, and overestimate the impact of assimilating sparse data, in this case BGC-Argo (Halliwell et al., 2014). This should be borne in mind when drawing conclusions.

## 4.2 Assimilation runs

For each of the five assimilation runs, profiles of global $MAE_{red\_\%}$ over FREE for December 2009 are plotted in Fig. 7, for nine model variables. The results for $MAE_{red\_abs}$ are very similar as for $MAE_{red\_\%}$ (not shown). For Chl-*a* (Fig. 7a), phytoplankton biomass (Fig. 7b), zooplankton biomass (Fig. 7c), and detrital nitrogen (Fig. 7d), which only have significant concentrations in the euphotic zone, profiles are plotted for the upper 250 m. For $NO_3$ (Fig. 7e), $O_2$ (Fig. 7f), DIC (Fig. 7g), alkalinity (Fig. 7h), and pH (Fig. 7i), profiles are plotted for the upper 2500 m. Recall that Chl-*a*, $NO_3$, $O_2$, and pH observations, assimilated in the BGC-Argo experiments, were produced for the upper 2000 m.

For Chl-*a*, OC, ARGO_1/4_OC, and ARGO_FULL_OC all had an $MAE_{red\_\%}$ value of 72 % at the surface. This suggests that ocean colour was very successful at improving surface Chl-*a*, while BGC-Argo was unable to add much information to that gained from the much denser ocean colour observations, at least at the global scale. When only BGC-Argo was assimilated, a small improvement at the surface of 7 % and 15 % was seen in ARGO_1/4 and ARGO_FULL respectively. Beneath the surface, at depths likely to see a deep chlorophyll maximum, BGC-Argo had much greater impact, with all four runs outperforming OC. ARGO_FULL outperformed ARGO_1/4, demonstrating benefit from extra in situ observations. Combining BGC-Argo and ocean colour gave better results at this depth in ARGO_1/4_OC (which is the proposed observing system), but ARGO_FULL and ARGO_FULL_OC performed similarly. Beneath the euphotic zone, where Chl-*a* was near-zero, the assimilation had little impact. Positive values of $MAE_{red\_\%}$ were seen below 250 m, but values of $MAE_{red\_abs}$ (not shown) tended to zero below about 220 m.

The results for phytoplankton biomass were very similar as for Chl-*a*. For zooplankton biomass, which was not directly updated by the assimilation, a large degradation in surface values was seen in all three runs assimilating ocean colour data, the impact reducing to near-zero at around 100 m. A smaller degradation was seen in ARGO_FULL, reduced further in ARGO_1/4. Detrital nitrogen was improved in the upper 100 m in all runs, especially those assimilating ocean colour, with little absolute impact of the assimilation beneath that depth.

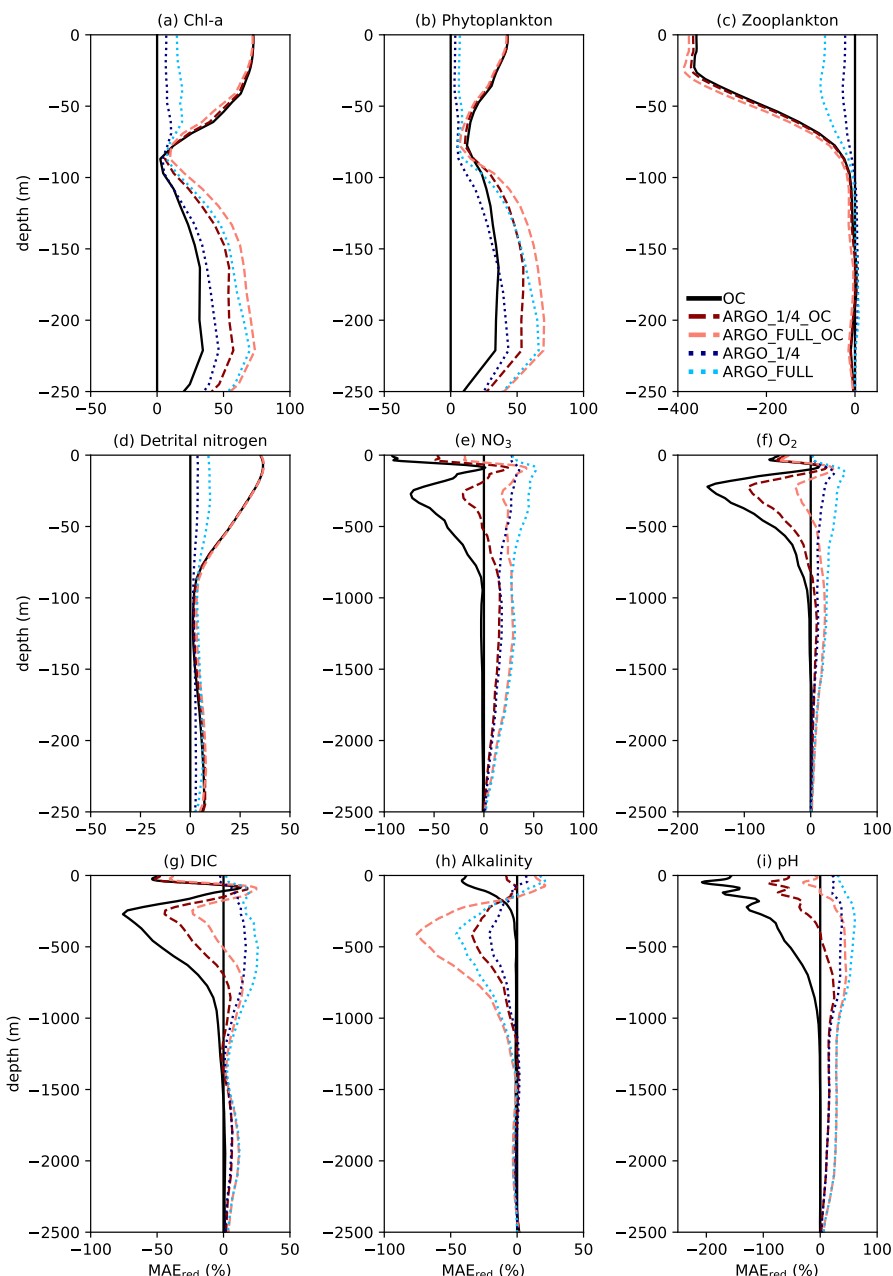

**Figure 7.** Profiles of global $MAE_{red\_\%}$ over FREE for December 2009. Note that axis scales differ between subplots.

For $NO_3$ and $O_2$, which were assimilated from the BGC-Argo floats, there was a clear improvement throughout the water
column to 2500 m in ARGO_1/4 and ARGO_FULL, with greater improvement when more floats were assimilated. Maximum
$MAE_{red\_\%}$ was seen at 100–120 m depth, with less impact at the surface, particularly for $O_2$. In OC, $NO_3$ and $O_2$ were degraded

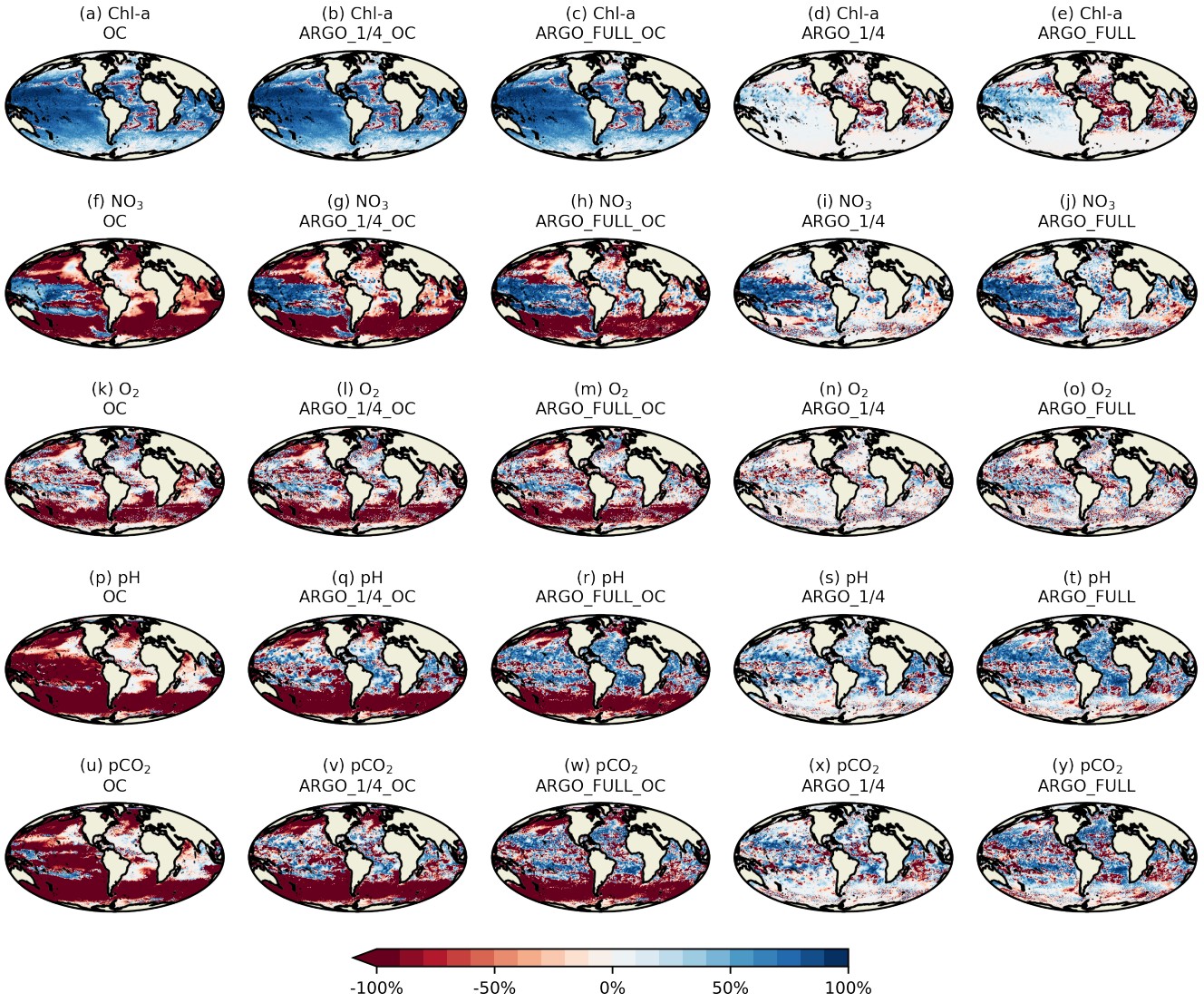

**Figure 8.** Surface $MAE_{red\_\%}$ over FREE for Chl-$a$, $NO_3$, $O_2$, pH, $pCO_2$ for December 2009.

in the upper 1000 m. Adding BGC-Argo to ocean colour assimilation partially mitigated this, with positive $MAE_{red\_\%}$ at some depths and negative $MAE_{red\_\%}$ at others.

With the carbon cycle, DIC, alkalinity, and pH were all degraded in OC. In ARGO_1/4 and ARGO_FULL, throughout most
445 of the water column DIC was improved and alkalinity degraded, with an overall improvement in the assimilated variable pH. Combining ocean colour and BGC-Argo assimilation gave mixed results, with a degradation in pH in the surface layers, but an improvement at depth.

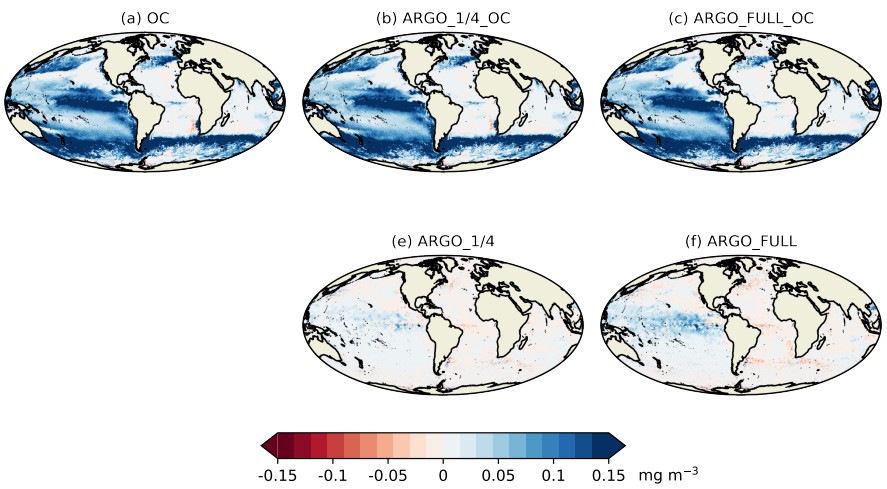

**Figure 9.** Surface $MAE_{red\_abs}$ over FREE for Chl-*a* for December 2009.

Spatial maps of surface $MAE_{red\_\%}$ over FREE for December 2009 are plotted in Fig. 8 for the five assimilation runs for Chl-*a*, $NO_3$, $O_2$, pH, and $pCO_2$. In OC surface Chl-*a* was almost universally improved (Fig. 8a), apart from a few small areas in the

Atlantic, North Pacific, and Indian Ocean. These correspond to areas where NATURE and FREE were almost identical, as seen in Fig. 3d, and so the observation errors were larger than the model error. Very little difference was made by adding BGC-Argo to ocean colour assimilation, while assimilating just BGC-Argo data gave mixed results for Chl-*a* at the surface. In the Pacific Ocean, Chl-*a* was slightly improved in ARGO_1/4 and further improved in ARGO_FULL, while in the Atlantic and Indian Oceans a degradation was seen. This again corresponds to regions where there was little absolute difference between NATURE

and FREE (Fig. 3d). As such, $MAE_{red\_\%}$ is not the most informative metric in these regions, and it is more appropriate to examine $MAE_{red\_abs}$. This is plotted for Chl-*a* in Fig. 9, which shows the absolute value of the degradation to be very small.

Surface $NO_3$ was degraded almost everywhere in OC (Fig. 8f), apart from the Tropical Pacific, which is where some of the largest differences were seen between NATURE and FREE (Fig. 3f). Adding BGC-Argo assimilation increased this improvement and started to reverse the degradation in other regions, particularly in ARGO_FULL_OC. Assimilating just BGC-Argo

improved $NO_3$ in most areas, with more impact seen with more floats, but the results were patchy in places.

The story for $O_2$ (Fig. 8k-o) was very similar as for $NO_3$, but with a smaller degradation introduced by ocean colour assimilation, and a smaller improvement brought about by BGC-Argo assimilation. For pH (Fig. 8p-t) and $pCO_2$ (Fig. 8u-y), the patterns were also similar, but with a greater degradation introduced by ocean colour assimilation, and a greater improvement brought about by BGC-Argo assimilation. Improvements to DIC and alkalinity when assimilating pH should also help improve

$pCO_2$, which was not directly assimilated, though the details of the impact differs between the two variables.

Current state-of-the-art reanalyses typically assimilate ocean colour data (Fennel et al., 2019), so to demonstrate the additional impact BGC-Argo might have in these systems, the remainder of this section focusses on $MAE_{red\_\%}$ over OC, rather than over FREE. Spatial plots of $MAE_{red\_\%}$ over OC for December 2009 at 100 m depth are shown in Fig. 10 for the assimilated

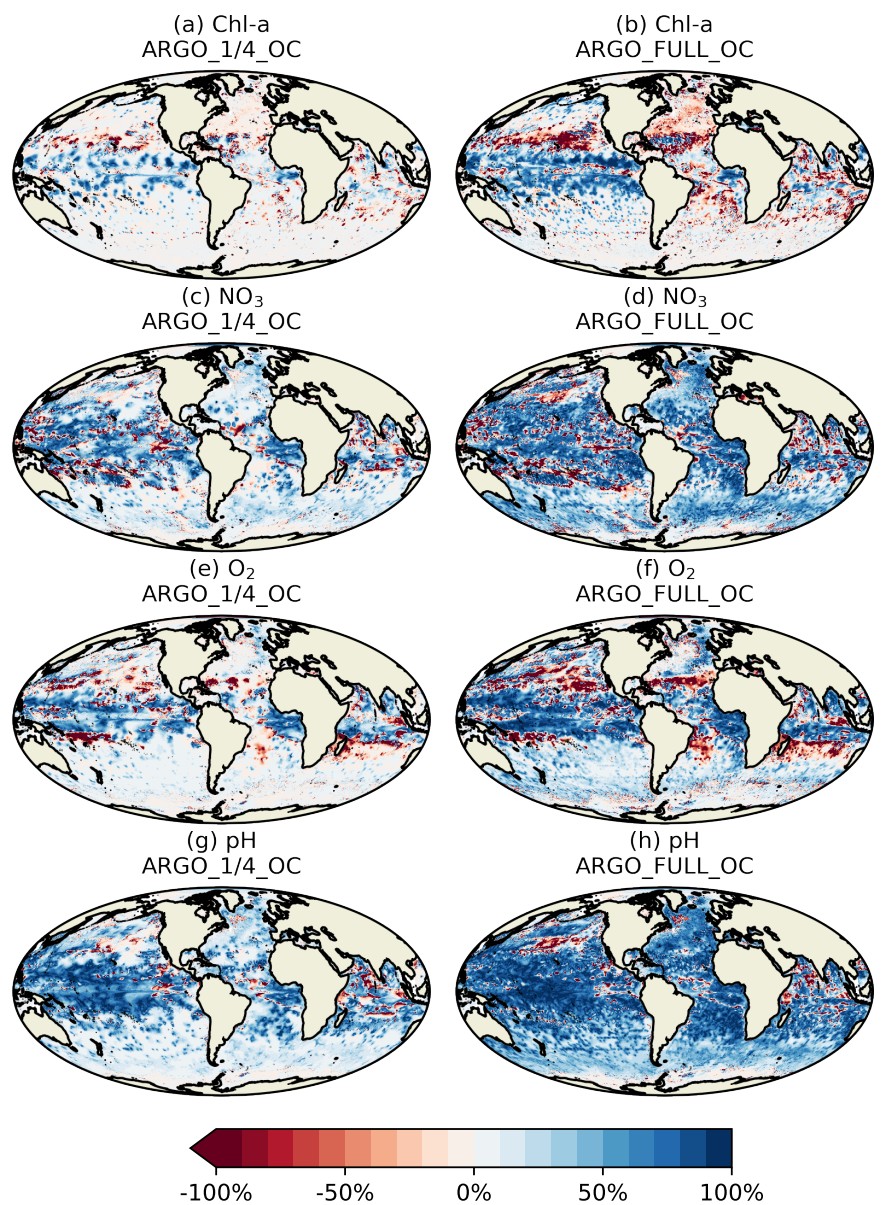

**Figure 10.** MAE$_{red\_\%}$ over OC for Chl-*a*, NO$_3$, O$_2$, pH for December 2009 at 100 m depth.

variables Chl-*a*, NO$_3$, O$_2$, and pH. Clear benefit was found for all variables, with greater improvements than at the surface. The
470 improvement was largest for pH and smallest for Chl-*a*, and in all cases a greater impact was seen in ARGO_FULL_OC than
ARGO_1/4_OC.

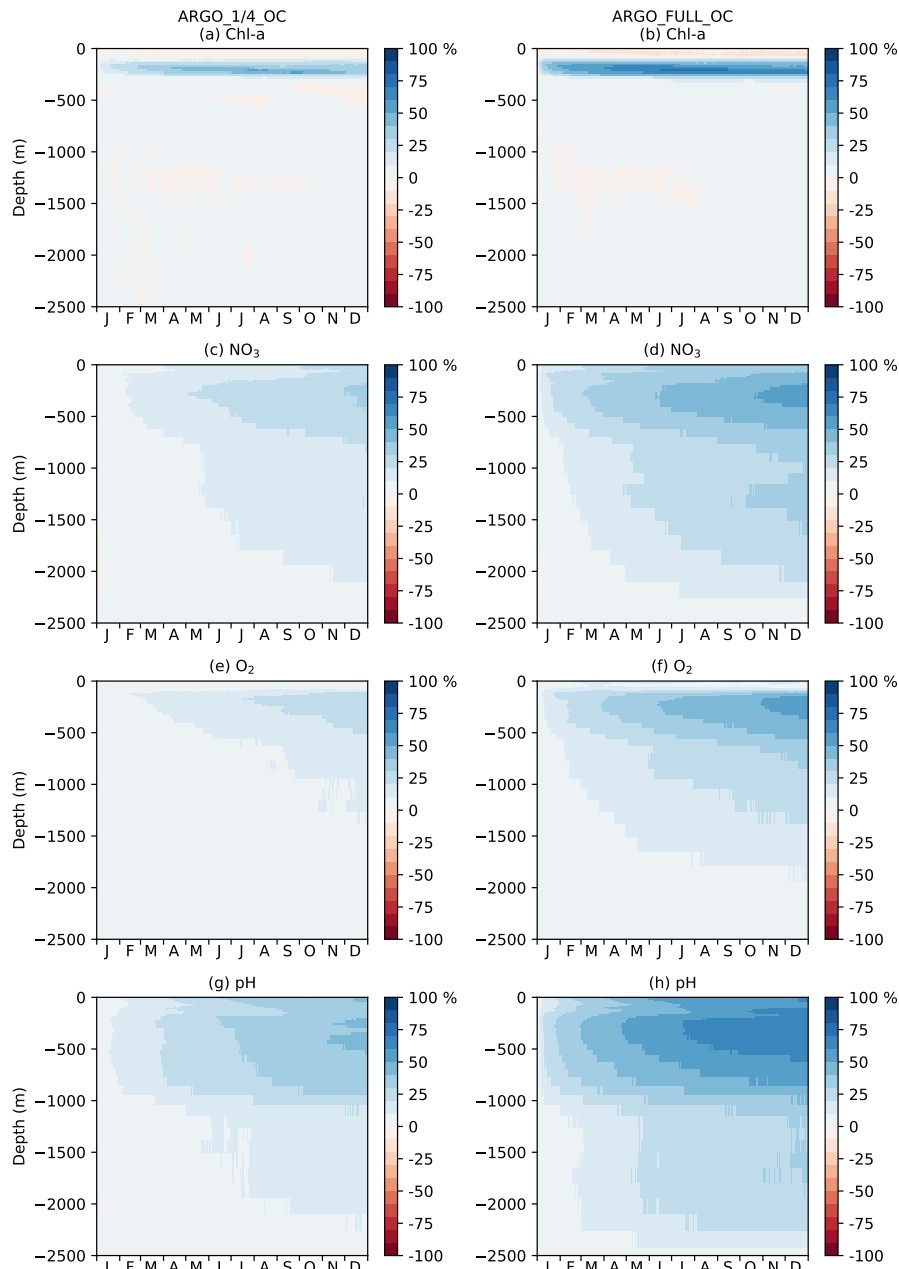

**Figure 11.** Hovmöller diagram of daily global $MAE_{red\_\%}$ over OC with depth for Chl-$a$, $NO_3$, $O_2$, and pH.

To investigate the impact of the BGC-Argo assimilation over the full year, a Hovmöller diagram (Hovmöller, 1949) of daily global $MAE_{red\_\%}$ over OC with depth is plotted for each of the assimilated variables, for ARGO_1/4_OC and ARGO_FULL_OC, in Fig. 11. For Chl-$a$ (Fig. 11a-b), ARGO_1/4_OC and ARGO_FULL_OC both displayed a very small degradation compared

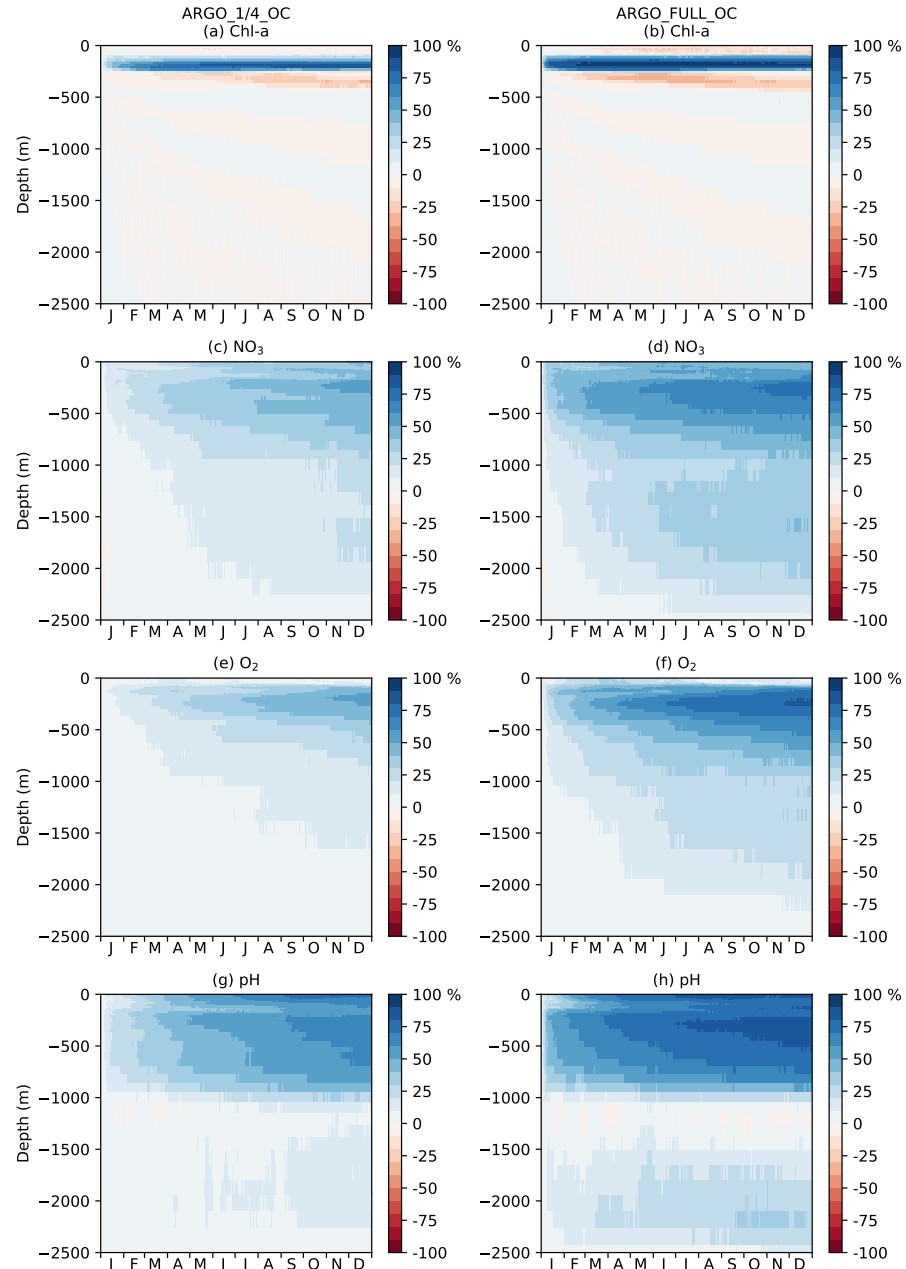

**Figure 12.** Hovmöller diagram of daily $MAE_{red\_\%}$ over OC with depth in the Tropical Pacific for Chl-*a*, $NO_3$, $O_2$, and pH.

with OC in the surface layers, throughout the year. This is consistent with the small difference seen between the runs in the profiles in Fig. 7a. Between about 80–300 m depth, where deep chlorophyll maxima may be found, a strong positive impact was found on Chl-*a*, strongest in ARGO_FULL_OC. In both runs this took a few weeks to spin up, then remained reasonably

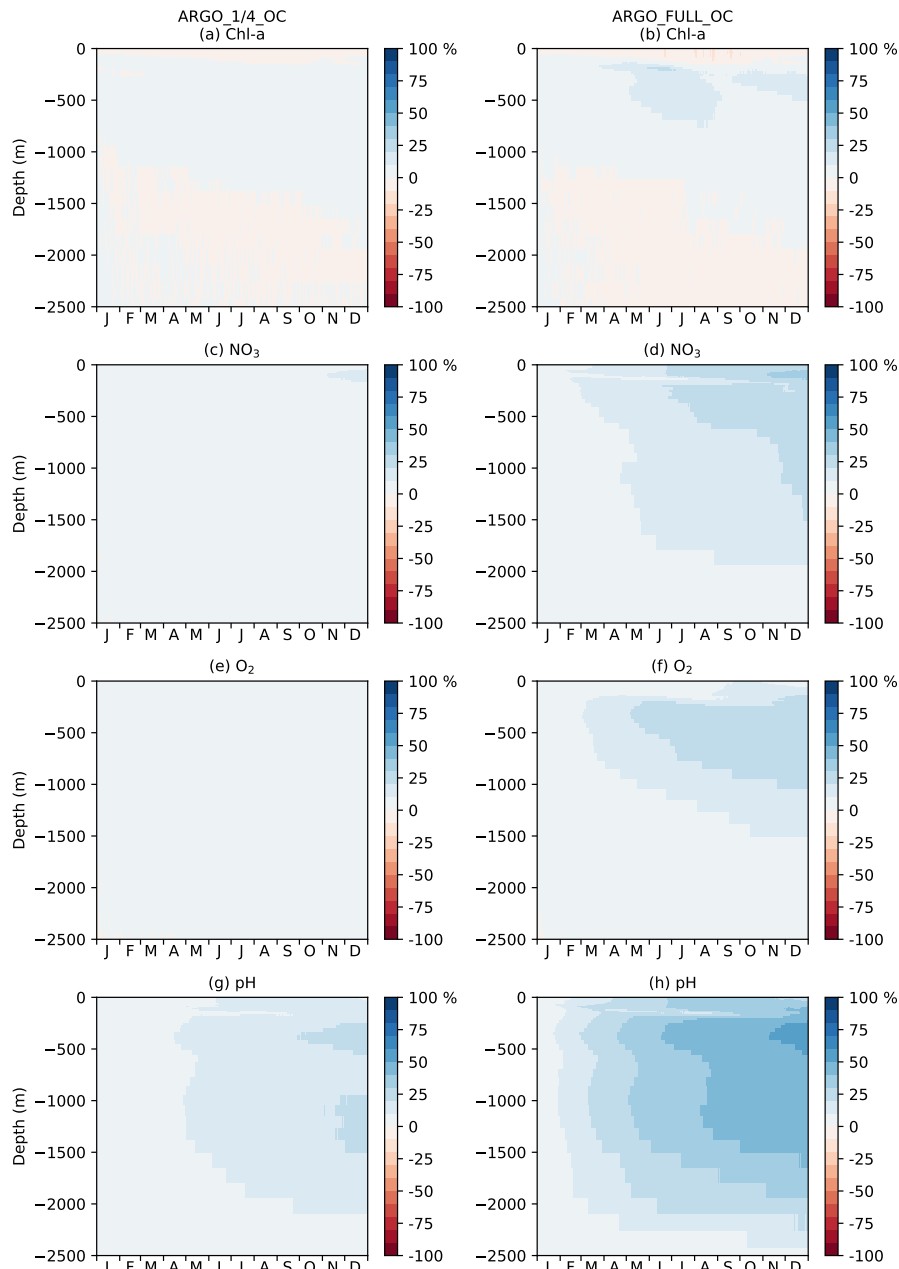

**Figure 13.** Hovmöller diagram of daily $MAE_{red}$ over OC with depth in the Southern Ocean for Chl-$a$, $NO_3$, $O_2$, and pH.

consistent throughout the year. Beneath about 300 m depth, values of $MAE_{red\_\%}$ were near-zero, as Chl-$a$ was negligible. For $NO_3$, $O_2$, and pH, an almost universal improvement was seen throughout the water column. For all three variables, values of $MAE_{red\_\%}$ were highest in the upper 1000 m, with a maximum at a similar depth or slightly deeper than that for Chl-$a$.

MAE$_{\text{red\_\%}}$ was consistently higher in ARGO_FULL_OC than in ARGO_1/4_OC, again demonstrating a positive impact from having a greater number of BGC-Argo floats. In both runs, MAE$_{\text{red\_\%}}$ increased throughout the year, suggesting that the impact of the assimilation was still spinning up, and the full potential would not be realised until the system had been run for longer than a year.

In Fig. 11, global MAE$_{\text{red\_\%}}$ is shown. As can be seen in Fig. 10, the sub-surface impact of the BGC-Argo assimilation was not globally uniform, with a stronger impact typically seen at low latitudes than at high latitudes. To demonstrate this further, equivalent versions of Fig. 11 are shown for the Tropical Pacific in Fig. 12, and for the Southern Ocean in Fig. 13. In the Tropical Pacific, the patterns for all the assimilated variables were very similar to the global average, but with higher MAE$_{\text{red\_\%}}$ values, showing a stronger positive impact of the assimilation. This was especially large for Chl-*a* between 80–300 m depth, and pH in the upper 1000 m. MAE$_{\text{red\_\%}}$ values for Chl-*a* were largely negative below 300 m, but MAE$_{\text{red\_abs}}$ values were near-zero (not shown), due to Chl-*a* concentrations being negligible. For NO$_3$, O$_2$, and pH, the impact of the assimilation was still spinning up after a year. In the Southern Ocean, MAE$_{\text{red\_\%}}$ values were much lower, with especially limited impact from the BGC-Argo assimilation in ARGO_1/4_OC. MAE$_{\text{red\_\%}}$ was typically largest for pH and lowest for Chl-*a*. The impact of the assimilation continued to increase with time, so may just have been taking longer to spin up than at lower latitudes. Similar results were seen in the Arctic Ocean (not shown). Increments also spread further in tropical regions due to longer correlation length-scales and faster propagation time-scales.

## 5  Summary and discussion

A set of observing system simulation experiments (OSSEs) was performed to explore the impact on global ocean biogeochemical reanalyses of assimilating currently-available ocean colour data, and assess the potential impact of assimilating BGC-Argo data. Two different potential BGC-Argo array distributions were tested: one where biogeochemical sensors are placed on all current Argo floats, and one where biogeochemical sensors are placed on a quarter of current Argo floats. This latter approximately corresponds to the proposed BGC-Argo array of 1000 floats (Roemmich et al., 2019). Assimilating the synthetic ocean colour data greatly improved the assimilated variable surface Chl-*a*, but had a mixed impact on the wider ecosystem and carbon cycle. Assimilating the synthetic BGC-Argo data gave no added benefit over ocean colour in terms of simulating surface Chl-*a*, but for most other variables, including sub-surface Chl-*a*, adding BGC-Argo improved results throughout the water column. This included surface pCO$_2$, which was not assimilated but is an important output of reanalyses. Both BGC-Argo array distributions gave benefits, with greater improvements seen with increased numbers of observations. The impact of the BGC-Argo assimilation was also increasing with time, and had not yet fully spun up after a year.

Real-world experiments assimilating Chl-*a* from ocean colour have widely found benefits when validating surface Chl-*a* against independent observations (Gehlen et al., 2015; Fennel et al., 2019), a conclusion echoed in this study. The impact of ocean colour assimilation on the wider model state has always been more ambiguous, with various studies reporting largely neutral or sometimes negative results, with some evidence of positive impacts highlighted (e.g. Gregg, 2008; Ciavatta et al., 2011; Fontana et al., 2013; Ford and Barciela, 2017). The sparsity of in situ observations, especially for variables such as

phytoplankton and zooplankton biomass, has always made it difficult to validate results, or compare conclusions from different studies. Many studies have used inherently multivariate assimilation methods such as the ensemble Kalman filter (Evensen, 2003), while others have employed balance relationships (Ford et al., 2012; Rousseaux and Gregg, 2012; Teruzzi et al., 2014; Skákala et al., 2018). This study used a form of the latter, with phytoplankton biomass variables updated to maintain background stoichiometric ratios. Updating phytoplankton biomass, a simple extra step which improved phytoplankton biomass itself and so should be expected to improve other model variables, resulted in a degradation of all other variables examined except for detritus. Zooplankton biomass was especially affected. It seems likely that this degradation occurred due to the changed MEDUSA parameter settings between NATURE and FREE, meaning that the underlying processes were altered such that identical concentrations of phytoplankton now led to different concentrations of other variables. Relevant perturbations included changes to the grazing half-saturation concentration for zooplankton, and nutrient uptake half-saturation concentrations for phytoplankton. This suggests that unless a given biogeochemical model can accurately describe all relevant biogeochemical processes in the ocean, which has not yet been demonstrated, simply improving phytoplankton concentrations might be as likely to degrade as improve other variables. This will also be the case for assimilation schemes which use ensembles to generate cross-correlations between Chl-$a$ and other model variables. These schemes are reliant on the model relationships between variables being correct, as it is these model relationships which the cross-correlations are based on. If the response of zooplankton to an increase in phytoplankton in the model ensemble differs from that in the real ocean, then the cross-correlations used in the assimilation will lead to a zooplankton response which follows the (incorrect) model rather than the real ocean, in exactly the same way as seen in this study. An alternative approach could be to use the nitrogen balancing scheme of Hemmings et al. (2008), which explicitly updates several model state variables to try and account for differing errors in phytoplankton growth and loss processes. This has been successfully used in previous ocean colour assimilation studies (Ford et al., 2012; Ford and Barciela, 2017; Ford, 2020) with the HadOCC model (Palmer and Totterdell, 2001). It was originally designed and tuned for use with HadOCC, so requires further development and tuning for use with the more complex MEDUSA, but an initial implementation for MEDUSA has been developed. Such a scheme offers more potential for controlling the wider biogeochemical state, especially if it could be expanded to alter parameter values as well as state variables.

Adding assimilation of BGC-Argo profiles of Chl-$a$, $NO_3$, $O_2$, and pH brought clear improvements to all assimilated variables, and some unassimilated ones. The impact was increased with a larger BGC-Argo array, suggesting that benefits may be seen up to and beyond the target array size of 1000 floats. The observations added important sub-surface information which cannot be obtained from satellite data, but which can yield improvements in simulations of variables such as air-sea $CO_2$ fluxes. All assimilated variables were greatly improved below the mixed layer. The greatest improvement was in terms of Chl-$a$ around the nitracline, at depths where deep chlorophyll maxima are likely to be found. This should help quantify a major contributor to global primary production, which cannot be observed from space. At the surface, more limited improvements were seen for Chl-$a$ and $O_2$, than for $NO_3$ and pH. This is likely due to the relative importance of different physical processes affecting these variables, and the density of data required for the assimilation to have a major impact. In the case of $NO_3$, and DIC which helps control pH, concentrations typically increase with depth, and the supply of $NO_3$ and DIC from below the mixed layer is a major contribution to surface values. Therefore, changes at depth due to the assimilation will alter surface values through

indirect processes. $O_2$ and Chl-*a* typically decrease in concentration with depth, and dynamics within the mixed layer are much more important in setting surface values. For $O_2$, major drivers are temperature and ocean–atmosphere exchange. For Chl-*a* a major driver is light availability. It seems that the BGC-Argo data was too sparse, even in ARGO_FULL, to have a widespread impact in these circumstances. More dedicated observations within the mixed layer may be likely to have more of an impact on surface values. For Chl-*a*, this can be successfully provided by ocean colour, as the results of this study show. For $O_2$ and other variables, alternative in situ observing technologies such as gliders may be able to play a role (Telszewski et al., 2018). It should be noted though that the OSSE framework used here did not consider possible real-world issues such as observation biases and inconsistencies between ocean colour and BGC-Argo Chl-*a* observations. Furthermore, for some variables and regions the error between the free-running model and the nature run was smaller than might be expected in real-world systems, potentially leading to an overestimate of the quantitative impact of BGC-Argo data (Halliwell et al., 2014).

For $NO_3$, $O_2$, and pH, the overall impact of the assimilation increased with time, and appeared to still be spinning up after a year. The likely explanation is that increments from individual BGC-Argo floats only influenced a relatively small local area, but this influence was long-lasting. Therefore, as further increments in different locations were added with time, the overall error of the system continued to decrease. This suggests that either the BGC-Argo observations will take a while to show their full benefit for assimilation, or a greater number of floats will be required, or changes to the assimilation method will be required to make better use of sparse observations. For Chl-*a*, which is typically more dynamic, this appeared to be less of an issue.

There is much scope for improving data assimilation methodologies to better use existing satellite data, and sparse in situ observations, which could bring at least as much benefit as expanding observing systems. Multivariate balancing, and better integration with physics data assimilation, may help improve unassimilated variables. More effective ways of spreading information from sparse data, such as cross-covariances based on empirical orthogonal functions or derived from an ensemble assimilation scheme, should also be considered. Related to this, the correlation length-scales used by the assimilation should be appropriately tuned for biogeochemical variables. In this study, a single horizontal correlation length-scale based on the first baroclinic Rossby radius was used, varying from a value of 25 km at low latitudes to 150 km at the Equator, following the physics implementation of Waters et al. (2015). This may help explain why the BGC-Argo assimilation demonstrated less widespread impact at high latitudes than near the Equator. A different correlation length-scale may be appropriate for biogeochemical variables. Furthermore, NEMOVAR has recently been developed to allow the use of multiple correlation length-scales (Mirouze et al., 2016), so both small- and large-scale corrections can be considered. The background error standard deviation estimates also need to be refined once real BGC-Argo observations are being assimilated, to reflect the background error in the assimilative system, which will depend on the actual distribution of BGC-Argo floats. The average ratio of background to observation error may also differ from that assumed in this study. It is likely that this would give different quantitative results, but the qualitative impact of the assimilation would remain similar.

A novel method for assimilating pH was introduced in this study, following the method for assimilating $pCO_2$ developed by While et al. (2012). The method corrects pH, a diagnostic variable in the model, by making the smallest combined change to DIC and alkalinity required to reach the target pH value. This was successful in improving both pH and DIC, but alkalinity was

often degraded. This highlights that making the smallest combined change to DIC and alkalinity does not guarantee that errors in both DIC and alkalinity are minimised. In some circumstances it might be more appropriate to make a smaller or no change to alkalinity, and a larger change to DIC. Or even to make a change of the opposite sign to alkalinity, and an even larger change to DIC to compensate. Unfortunately, without concurrent observations of DIC or alkalinity, this information is not known at the time of assimilation. This is equally the case whether $pCO_2$ or pH is being assimilated, and so it was decided that the safest assumption would be to make the smallest combined change to DIC and alkalinity. In light of these results it may be worth revisiting this assumption.

From the point of view of ocean data assimilation, BGC-Argo will bring significant advances in reanalysis and forecasting skill, and it is recommended to proceed with its development as a priority. The proposed array of 1000 floats will be enough to deliver clear improvements, and a larger array would be likely to bring further benefits still. Ocean colour and BGC-Argo provide complementary information, so maintaining and developing the existing ocean colour satellite constellation should also be a priority. Technologies such as gliders may also bring additional benefits, for instance for $O_2$ in the mixed layer.

*Data availability.* The nature of the 4D data generated in running the model experiments requires a large tape storage facility. These data are in excess of 100 terabytes (TB). However, the data can be made available upon request from the author.

*Competing interests.* The author declares that they have no conflict of interest.

*Acknowledgements.* The author would like to thank Susan Kay and Matt Martin for useful discussions and comments on the draft manuscript, as well as the two anonymous reviewers for their helpful comments in the Biogeosciences discussion forum. This study used synthetic observations based on Argo data. These data were collected and made freely available by the International Argo Program and the national programs that contribute to it. (http://www.argo.ucsd.edu, http://argo.jcommops.org). The Argo Program is part of the Global Ocean Observing System. This study also made use of data from the Ocean Colour CCI, GLODAP, World Ocean Atlas, and EN4 projects, as well as the $pCO_2$ analysis product of Landschützer et al. (2015a, b). The author would like to thank all those involved in collecting, processing, and distributing these data, and making them available for public use.

*Financial support.* This study received funding from the European Union's Horizon 2020 Research and Innovation program under Grant Agreement 633211 (AtlantOS).

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
