# Peer review of "Assimilating synthetic Biogeochemical-Argo and ocean colour observations into a global ocean model to inform observing system design"

_Biogeosciences, 2020_

## Referee Comment (RC1) · Anonymous Referee #1 · 3 Jun 2020

**General comments:**

This study tackles an important question regarding the potential benefit of assimilating BGC-Argo profiles in improving the global ocean biogeochemical reanalyses. The manuscript consists of 3 sections: 1) establish an OSSEs framework; 2) assess different strategies of updating BGC model state variables when assimilating ocean color data; 3) evaluate the benefit of assimilating different numbers of BGC-Argo profiles in addition to ocean color data over the assimilation of profiles or ocean color data alone. Overall, I think this work is well-conceived and will make an important contribution to the state-of-art ocean biogeochemical data assimilation combining the routinely available ocean color data and the emerging BGC-Argo observations.

I have two major comments. First, I feel that the second section of the manuscript on different DA strategies, at its present form, does not add much value to the story and the selection of best DA strategy involving a nitrogen balancing scheme before thorough tuning doesn't seem fair. Second, the first section on OSSEs requires a bit more analysis to prove its credibility. I'll provide more detailed explanations below. Aside from that, I have some minor comments, mostly technical, for the author to consider.

Upon appropriately addressing these comments, I'll recommend publication of the manuscript in Biogeosciences.

1. I question on the value of including section 2 on comparing different update strategies when assimilating surface chl data for following reasons:

1) While I acknowledge the efforts and time needed for comparing 6 different DA strategies, I feel that the present comparison is not sufficient for fairly selecting the best DA strategy. I would argue that the more sophisticated nitrogen balancing scheme failing to outperform other strategies is largely because the parameter values used in the scheme are directly adopted from Hemmings et al. (2008) without proper tuning. These parameters reflect the BGC model's inherent relationships between chl and other model state variables. Since the model used here (MEDUSA) has quite different structure from that of Hemmings (HadoCC), a careful calibration of the parameters in the N balancing scheme is needed before its usage. That maybe contribute to a separate manuscript focusing on the benefit of multivariate BGC update over single-variable update.

2) At its present form, I didn't see strong connection between section two and three in the manuscript. To me, the most significant findings are from section three and this section stands out even if section two is completely removed. This is because the comparison in section two didn't suggest a clear winner and the ultimate decision of using the DA strategy of an intermediate complexity rather than the most sophisticated one (the N balancing

scheme) for section three further reduce the value of including the entire comparison in section two.

3) If section two was removed, the author can have more space to elaborate and focus on the impacts of assimilating BGC-Argo profiles on different variables and suggesting directions for future work to improve. Currently, I feel the discussion on this part is relatively short compared to the emphasis it receives in the title, abstract and Introduction.

2. I think section one on establishing the OSSEs framework is key to the credibility of assessment on assimilation impact. Presently the only analysis provided to show the credibility of OSSEs is a comparison of the errors between FREE and OBS and between FREE and NATURE for surface chl, NO3 and pCO2 in Figure 2. According to the criteria of designing rigorous ocean OSSE system detailed in Halliwell et al. (2014), I would request the author to comment and/or provide some information on following aspects:

1) Can the NATURE run reasonably capture the key features measured by the observing systems (in this case the surface chla, and the BGC profiles)? The author refers the performance of NATURE run to references given in Section 2 which is not very clear to me which one exactly has the same configuration and time period as the one here. A brief summary and/or some figures on the performance of the NATURE run will help.

2) Figure 2 only presents the surface comparisons. Since the assessment involves the vertical profiles, can the author also comment on whether the errors between FREE and NATURE are comparable to those between FREE and OBS in terms of the vertical distribution pattern of observable BGC variables.

3) How about the error growth rate? One important criterion of credible OSSE evaluation is that the differences between the FREE and the NATURE ("truth") grow at the same rate as errors that develop between the state-of-the-art ocean models and the true ocean (Halliwell et al. 2014).

Halliwell, G. R., Srinivasan, A., Kourafalou, V., Yang, H., Willey, D., Le Hénaff, M., and Atlas, R.: Rigorous evaluation of a fraternal twin ocean OSSE system for the open Gulf of Mexico, J. Atmos. Ocean Tech., 31, 105–130, https://doi.org/10.1175/JTECH-D-13-00011.1, 2014.

**Specific comments:**

L21: '… half the planet's primary production.' Reference?

L75-77: Could you briefly add the outcome of assimilating these BGC-Argo observations in these two studies?

L87-88: 'Two groups perform … and the Met Office … presented here.' this information may be meaningful for the groups involved but doesn't seem informative for general readers. Do the two groups aim at different perspectives of the BGC assessment? What are they then?

L116-118: Isn't that oxygen and dissolved inorganic N are also simulated? Or are they implicitly included in the 'coupled carbon cycle'? It's not clear to me what 'coupled' means here.

L154: This is acceptable, but could you comment if the physics of the NATURE run without DA is reliable to conduct the OSSEs?

L156: Just curious if there is any particular reason for using log10 instead of log-normal transformation.

L160: How large is the correlation length-scale? Water et al. (2015) is on physical DA. Same length scale used for the BGC assimilation here? I'm thinking that BGC fields are more dynamic and thus have a smaller correlation length-scale.

L165-170: Can the surface information help constrain the BGC fields below the mixed layer? '… below the mixed layer the vertical length-scale increases with the model's vertical grid resolution.' this is confusing to me.

L172: 'The increments … from the two methods should be similar, though not identical.' Why? Isn't that the two methods have different treatments below the mixed layer?

L191: are these ratios fixed or time-dependent?

L216-217: this sentence should be reworded, something like: 'The approach taken to the assimilation of partial pressure of $CO_2$ (pCO2) into HadOCC (While et al., 2012) is therefore adopted here with pH. In HadOCC, pCO2 is a function of temperature, …'

L261: Fujii et al. 2019 suggested the assimilative model to be configured either in reduced resolution or sufficiently different physical parameterizations.

L272: 'year 5000', is it true or typo?

L311: 30% is fine for estuarine and coastal waters, but would it be too large for chl-a profiles in open ocean?

L327: for these variables, are the error standard deviations fixed or monthly varying as well?

L357 & Table 1: would it be clearer to reserve the term 'control run' for the definition in Eq 2 only and call the 'non-assimilative run' the 'free run' throughout the text?

L391: What's the DA impact for depths below 250 m?

Figure 7 does not include O2 or pH while Figure 8 does. What's the rationale of presenting different set of variables here?

L522: Any comment on why O2 is not improved by BGC-Argo data? And why "in situ technologies such as gliders" can play a role?

---

## Referee Comment (RC2) · Anonymous Referee #2 · 14 Jun 2020

The manuscript by Ford presents an OSSE experiment to investigate a number of assimilation strategies for ocean colour (OC) and biogeochemical Argo (BGC-Argo) observations using an already published DA method. The simulations are performed using a global model and sets of synthetic observations that resemble the current L3 chlorophyll OC and two potential arrays of BGC-Argo based on the current Argo network.

The manuscript is well written and the performed modelling experiments allow novel and useful insights on the integration of BGC-Argo and OC data into global model assimilation. However, as presented, results seem rather superficial. The work would be

the basis for a very valuable paper, but the results need to be explored further before I can recommend publication. Âă My main concern is that results present a single month of simulation (i.e., the last month of a 2-year simulation: 1-year spinup and 1-year assimilation) and a single global statistics (e.g., Figs. 3, 4 and 10). Conclusion/discussion on assimilation strategies and impact of the observing systems are possibly misled by the limited results. Results of the whole year of the assimilation runs should be presented and the MEAred maps and profiles enriched with spatial and temporal statistics to provide quantitative insights on the impact of BGC-Argo in different seasons and regions of the global ocean. In particular, which are the areas and seasons that could benefit most from BGC-Argo assimilation and the integration of the two observing systems? I feel that the manuscript misses the objective to provide useful indications to design future observing system strategy, as it is proposed in the title.

A second issue concerns the comparison between the PHY (phytoplankton update) and NIT (nitrogen balancing scheme) assimilation schemes. While the novelty of the OSSE experiment is related to the integration of OC and BGC Argo, the lack of the NIT implementation for BGC-Argo assimilation is a significant limitation of the manuscript that should be discussed. L425-427 are misleading. In fact, the choice of the PHY update scheme is explained at L229-230 (i.e., apparently, the NIT method has not been implemented for the joint BGC-Argo and OC assimilation). The joint OC and BGC-Argo assimilation strategy should be clearly described at L347-348. Then, it is not clear the objective of the first set of experiments (OC assimilation, which conclusions are mainly already published) if its results are not used for the second set of experiments. Even if the NIT method has not been implemented, the manuscript can be completed by a more unbiased discussion on pro and cons of the two methods and by presenting the work to be done and the the benefits to have the NIT method working with the BGC-Argo assimilation. In fact, some conclusions seem misleading. While the nitrogen balancing scheme is reported as the method with more potentiality (L406-407 in results and L510-L515 in discussion), results on BGC-Argo assimilation runs do not support this conclusion. I suggest to clarify better the proposed assimilation strategy and to

detail better what would be needed to have the NIT method working with BGC-Argo assimilation. The conclusion that "only minimally altered for use with MEDUSA, so more specific tuning may help (L516-517)" seems misleading.

Minor points:

Line 173: why should the two methods provide similar increments? One is uniform with depth from surface to MLD depth, while the second method is not limited to the MLD and vertical increments are mediated by covariance.

Line 335: can the author provide some more details on how the observation and background errors have been matched? and which is the value of inflation?

Line 336: this sentence is not clear: the observation error is set from the real global BGC-float array, so what will change when the system is functioning with real BGC-floats? More generally, the discussion missed to tackle this topic: how much are the results affected by the selected observation and background errors? would it be a different impact of the two observing systems using different observation errors?

Table 3 can be enriched to improve the identification of the runs at a glance. I would suggest to add 2 new columns for the assimilated and updated variables, and to split the note column in two new ones: background error (i.e., vertical propagations: 2D and 3D) and type of increment.

L354-355: explain how MAEosse and MAEcontrol are computed for pixels in the maps of Figures 5-9 and for points of the profiles in Figures 3, 4 and 10. Which are the distribution compared?

L377: the absolute differences in Atlantic and Indian Oceans appear significant (i.e., 1 order magnitude). Can the author provide more details about which modifications of the FREE run (w.r.t. NATURE run) should have served to increase nutrient concentration? And which modifications compensate it (L382)? I agree that achieving a global appropriate level of error with a complex model (with uniform parameterization) can not

be managed, however the author should provide some details on which modifications didn't work as expected. This can be helpful in understanding the effectiveness of the data assimilation in those areas.

L385: the absolute difference of NO3 and pCO2 between FREE and NATURE seems much lower than that between FREE and real observation. Can the author discuss which are the implications of the low difference for the OSSE assimilation results? I would argue that the effectiveness of the assimilation might be limited in some areas by the low differences between the synthetic observations generated from NATURE and the FREE run. Further, I would argue that MAEred might not be a good metric because of the MAEcontrol at the denominator in the areas where NATURE and FREE are so close.

L395: Since large areas of the global ocean are characterized by a DCM below 60m depth, the degradation of MAE below 60m would deserve a more detailed comment.

L405: why should the similar behaviours of OC_2D and OC_3D demonstrate that the use of NEMOVAR to create 3D increments for the combined OC and BGC-Argo float assimilation is fit-for-purpose?

Figure 4: I would suggest to reduce y-axis to 0-250m depth for the chlorophyll, phytoplankton and zooplankton plots to increase their readability. Are the high positive values below 250 in chla and phytoplankton plots due to the very low values of those variables below the euphotic zone?

L450-451 and L477-478: Can the degradation of Alkalinity be due to an improper working of the smallest combined change of DIC and Alk with pH assimilation? A more detailed analysis is expected to show the pro and cons of the method when pH is used instead of pCO2.

L454-L455: why are the negative values of MAEred in the Atlantic and Indian Oceans (Fig. 5) related to the compensating errors introduced in FREE? Since the FREE and

NATURE differences are very low in those areas (Fig. 2), the impact of the assimilating synthetic observations (generated from NATURE) should be negligible. I wonder whether the MAEred is not a good metric because of the MAEcontrol at the denominator for those areas.

Figures 8 and 10 seem redundant and not necessary. For example, the MAEred over OC_3D_PHY of ARGO_FULL_OCÂă(Fig. 8a and Fig 10a) provides the same information (except for the normalization of denominator) of the difference between MAEred over FREE of ARGO_FULL_OC and MAEred over FREE of OC_3D_PHY (Fig. 4a and Fig 5a and c). I suggest that the relative impact of adding BGC-Argo can be shown by a new table of the MAEred over FREE numeric values. The table can report values for selected regions and different seasons/months providing indications of which areas of the global ocean and periods of the year can benefit most by the BGC-Argo assimilation.

L505-508: which parameter settings between NATURE and FREE caused the degradation of the other variables? Can additional details be added?

L510-511 please explain: this seems not supported by results or references.

L522: BGC-Argo assimilation has a small and positive impact on O2 as shown in Figure 10f (BGC-Argo assimilation). The degradation of O2 at surface seems due to OC assimilation (see figure 4f).

L531-L535: DA method improvements are important, but the paper does not really tackle this aspect; thus, those lines may fit better in the introduction and not as the last conclusion.

---

## Author Comment (AC1) · 31 Jul 2020

**General comments:**
**This study tackles an important question regarding the potential benefit of assimilating BGC-Argo profiles in improving the global ocean biogeochemical reanalyses. The manuscript consists of 3 sections: 1) establish an OSSEs framework; 2) assess different strategies of updating BGC model state variables when assimilating ocean color data; 3) evaluate the benefit of assimilating different numbers of BGC-Argo profiles in addition to ocean color data over the assimilation of profiles or ocean color data alone. Overall, I think this work is well-conceived and will make an important contribution to the state-of-art ocean biogeochemical data assimilation combining the routinely available ocean color data and the emerging BGC-Argo observations.**

Thank you for your positive assessment and constructive comments. I will address each of these in turn below.

**I have two major comments. First, I feel that the second section of the manuscript on different DA strategies, at its present form, does not add much value to the story and the selection of best DA strategy involving a nitrogen balancing scheme before thorough tuning doesn't seem fair. Second, the first section on OSSEs requires a bit more analysis to prove its credibility. I'll provide more detailed explanations below. Aside from that, I have some minor comments, mostly technical, for the author to consider.**

Thank you for these comments. On reflection, I agree that the section on ocean colour assimilation strategies does not add much value to the core aims of the manuscript, and is not fully fledged. I therefore propose to remove this section, just presenting the results of the OC_3D_PHY run alongside the BGC-Argo assimilation. I will potentially develop the ocean colour assimilation work further in a future publication.

I also agree that presenting further analysis of the OSSE framework would help demonstrate its credibility, and propose to include the types of assessment requested below.

**Upon appropriately addressing these comments, I'll recommend publication of the manuscript in Biogeosciences.**

**1. I question on the value of including section 2 on comparing different update strategies when assimilating surface chl data for following reasons:**

**1) While I acknowledge the efforts and time needed for comparing 6 different DA strategies, I feel that the present comparison is not sufficient for fairly selecting the best DA strategy. I would argue that the more sophisticated nitrogen balancing scheme failing to outperform other strategies is largely because the parameter values used in the scheme are directly adopted from Hemmings et al. (2008) without proper tuning. These parameters reflect the BGC model's inherent relationships between chl and other model state variables. Since the model used here (MEDUSA) has quite different structure from that of Hemmings (HadoCC), a careful calibration of the parameters in the N balancing scheme is needed before its usage. That maybe contribute to a separate manuscript focusing on the benefit of multivariate BGC update over single-variable update.**

I agree with this assessment. Recalibrating the parameterisations of the Hemmings et al. (2008) nitrogen balancing scheme for specific use with MEDUSA would be a considerable amount of work, but necessary to achieve the best results. As suggested, I will therefore

remove this section from the current manuscript, and potentially develop the work further as part of a separate paper.

**2) At its present form, I didn't see strong connection between section two and three in the manuscript. To me, the most significant findings are from section three and this section stands out even if section two is completely removed. This is because the comparison in section two didn't suggest a clear winner and the ultimate decision of using the DA strategy of an intermediate complexity rather than the most sophisticated one (the N balancing scheme) for section three further reduce the value of including the entire comparison in section two.**

I agree that the most significant findings, in relation to the core aims of the study, are in the final section, and that this section deserves the most attention.

**3) If section two was removed, the author can have more space to elaborate and focus on the impacts of assimilating BGC-Argo profiles on different variables and suggesting directions for future work to improve. Currently, I feel the discussion on this part is relatively short compared to the emphasis it receives in the title, abstract and Introduction.**

I agree. I therefore propose to remove the ocean colour section, and use the extra space to further develop the assessment and discussion surrounding BGC-Argo assimilation, as suggested.

**2. I think section one on establishing the OSSEs framework is key to the credibility of assessment on assimilation impact. Presently the only analysis provided to show the credibility of OSSEs is a comparison of the errors between FREE and OBS and between FREE and NATURE for surface chl, NO3 and pCO2 in Figure 2. According to the criteria of designing rigorous ocean OSSE system detailed in Halliwell et al. (2014), I would request the author to comment and/or provide some information on following aspects:**

**1) Can the NATURE run reasonably capture the key features measured by the observing systems (in this case the surface chla, and the BGC profiles)? The author refers the performance of NATURE run to references given in Section 2 which is not very clear to me which one exactly has the same configuration and time period as the one here. A brief summary and/or some figures on the performance of the NATURE run will help.**

The references in Section 2 detail assessment of the model components used, but not of the specific NATURE run which is newly presented in this manuscript. I will be clearer about this in the text. I will also present some figures and assessment in the revised manuscript demonstrating the performance of the NATURE run as requested. I can confirm that it is able to reasonably capture key features, but agree that this isn't currently shown in the manuscript and should be.

**2) Figure 2 only presents the surface comparisons. Since the assessment involves the vertical profiles, can the author also comment on whether the errors between FREE and NATURE are comparable to those between FREE and OBS in terms of the vertical distribution pattern of observable BGC variables.**

I will present further assessment in the revised manuscript comparing the errors between FREE and NATURE and FREE and OBS in terms of vertical distribution for biogeochemical variables for which there are appropriate observation products for the comparison.

**3) How about the error growth rate? One important criterion of credible OSSE evaluation is that the differences between the FREE and the NATURE ("truth") grow at the same rate as errors that develop between the state-of-the-art ocean models and the true ocean (Halliwell et al. 2014).**

I will also present further assessment of the error growth rate, I agree this is important to demonstrate.

**Halliwell, G. R., Srinivasan, A., Kourafalou, V., Yang, H., Willey, D., Le Hénaff, M., and Atlas, R.: Rigorous evaluation of a fraternal twin ocean OSSE system for the open Gulf of Mexico, J. Atmos. Ocean Tech., 31, 105–130, https://doi.org/10.1175/JTECH-D-13-00011.1, 2014.**

**Specific comments:**

**L21: '… half the planet's primary production.' Reference?**

I will add a reference to Field et al. (1998, http://doi.org/10.1126/science.281.5374.237).

**L75-77: Could you briefly add the outcome of assimilating these BGC-Argo observations in these two studies?**

I will add details of these. Verdy and Mazloff (2017) produced a five-year state estimate of the Southern Ocean using an adjoint method, and were able to capture over 60% of the variance in oxygen profiles at 200 m and 1000 m depth. Cossarini et al. (2019) assimilated chl-a into a model of the Mediterranean Sea, and found this was successful in adjusting the shape of chlorophyll profiles, and that with the present number of BGC-Argo floats they could constrain phytoplankton dynamics in up to 10% of the Mediterranean Sea.

**L87-88: 'Two groups perform … and the Met Office … presented here.' this information may be meaningful for the groups involved but doesn't seem informative for general readers. Do the two groups aim at different perspectives of the BGC assessment? What are they then?**

I will rephrase this section to be less focussed on the details of the project, and instead give a brief summary of the results of Germineaud et al. (2019), who presented a probabilistic evaluation at a single assimilation time step, finding that chl-a from BGC-Argo floats added value at surface locations where ocean colour was unavailable, and at depth.

**L116-118: Isn't that oxygen and dissolved inorganic N are also simulated? Or are they implicitly included in the 'coupled carbon cycle'? It's not clear to me what 'coupled' means here.**

This sentence was poorly phrased and incomplete. Oxygen and DIN are indeed also simulated, as are iron, silicate, and fast- and slow-sinking detritus. I will rephrase the sentence to give a more complete description of the model variables. The word "coupled" is unnecessary in this context, and I will remove it. The model DIC and alkalinity are the state variables representing the carbon cycle.

**L154: This is acceptable, but could you comment if the physics of the NATURE run without DA is reliable to conduct the OSSEs?**

The NATURE run is able to capture key features of the physics, I will present assessment of this alongside the similar assessment of biogeochemical variables discussed above.

**L156: Just curious if there is any particular reason for using log10 instead of log-normal transformation.**

In practice, it should make no difference to the assimilation whether log-normal or $\log_{10}$ is used. The shape of the distribution is the same (the ratio of $\log(x)$ to $\log_{10}(x)$ is identical for all values of x), except that $\log_{10}$ gives a smaller variance. It is the shape of the distribution that matters for the assimilation, so as long as the same transformation is applied consistently to both model and observations, it should not matter whether log-normal or $\log_{10}$ is used. In the literature, some studies use $\log_{10}$ (e.g. Gregg, 2008, https://doi.org/10.1016/j.jmarsys.2006.02.015), while others use log-normal (e.g. Ciavatta et al., 2011, https://doi.org/10.1029/2011JC007219). The decision to use $\log_{10}$ here is a historical one, following Ford et al. (2012, https://doi.org/10.5194/os-8-751-2012), but as stated the choice should make no difference.

**L160: How large is the correlation length-scale? Water et al. (2015) is on physical DA. Same length scale used for the BGC assimilation here? I'm thinking that BGC fields are more dynamic and thus have a smaller correlation length-scale.**

The correlation length-scale is the same as in Waters et al. (2015, https://doi.org/10.1002/qj.2388), and varies with the Rossby radius, from a value of 25 km at low latitudes to 150 km at the Equator (see Fig. 3 of Waters et al., 2015, https://doi.org/10.1002/qj.2388). For a 1/4° resolution ocean model, which is limited in its resolution of mesoscale features, using the same correlation length-scale for BGC as for physics is probably appropriate for an initial implementation. It is true though that the appropriate correlation length-scale(s) to use for assimilating BGC-Argo is an open question, and this should be addressed in future development of the assimilation. I will therefore clarify in the text the value of the length-scale used, and add some discussion of these issues to the final Discussion section.

**L165-170: Can the surface information help constrain the BGC fields below the mixed layer? '… below the mixed layer the vertical length-scale increases with the model's vertical grid resolution.' this is confusing to me.**

I will rephrase and expand the description of the vertical correlation length-scales to try to be clearer. The length-scales are designed to limit the spreading of information across the base of the mixed layer. For an example see Fig. 4 of Waters et al. (2015, https://doi.org/10.1002/qj.2388) and the surrounding description in their Section 3.6.

For surface observations the length-scale is set equal to the mixed layer depth, meaning that information from the surface observations is spread to the base of the mixed layer, but has limited impact on BGC fields below it. This is a deliberate decision based on the lack of correlation between water mass properties in and below the mixed layer. This is likely to be the case as much for BGC as for physics (see e.g. Fontana et al., 2013, https://doi.org/10.5194/os-9-37-2013).

The vertical correlation length-scale is set to a minimum value at the mixed layer depth, and then increases with depth (see Fig. 4 of Waters et al., 2015, https://doi.org/10.1002/qj.2388). This increase is proportional to the increase in vertical model grid spacing that occurs with depth. I will rephrase the description in the text to make this clearer.

**L172: 'The increments … from the two methods should be similar, though not identical.' Why? Isn't that the two methods have different treatments below the mixed layer?**

As I propose to remove the section comparing ocean colour assimilation strategies, I will also remove this sentence.

**L191: are these ratios fixed or time-dependent?**

Again, as I propose to remove the section comparing ocean colour assimilation strategies, I will also remove this section. But to answer the question, the ratios are time-dependent, based on the background ratios in each assimilation cycle.

**L216-217: this sentence should be reworded, something like: 'The approach taken to the assimilation of partial pressure of CO2 (pCO2) into HadOCC (While et al., 2012) is therefore adopted here with pH. In HadOCC, pCO2 is a function of temperature, …'**

I will reword the sentence as suggested.

**L261: Fujii et al. 2019 suggested the assimilative model to be configured either in reduced resolution or sufficiently different physical parameterizations.**

I will clarify this in the text.

**L272: 'year 5000', is it true or typo?**

This is true. Spinning up UKESM1 for CMIP6 was a massive endeavour, as documented by Yool et al. (2020, https://doi.org/10.1029/2019MS001933).

**L311: 30% is fine for estuarine and coastal waters, but would it be too large for chl-a profiles in open ocean?**

30% is a commonly used value in open ocean chlorophyll assimilation studies (e.g. Pradhan et al., 2020, https://doi.org/10.1029/2019MS001933), and especially for daily products I would expect this to be appropriate. For instance, Krasemann et al. (2017, https://esa-oceancolour-cci.org/sites/esa-oceancolour-cci.org/alfresco.php?file=6d534e45-fbfd-4cc5-8125-d84f0b3abea6&name=OC-CCI-PVIR-v3.20170303.pdf) found an RMSD of 0.31 for ocean colour matchups of $\log_{10}$(chl-a) against in situ observations. Maritorena et al. (2010, https://doi.org/10.1016/j.rse.2010.04.002, Fig. 10) estimated errors to be in excess of 30% across much of the ocean for daily products, though lower for monthly composites.

**L327: for these variables, are the error standard deviations fixed or monthly varying as well?**

They are fixed, I will make this clearer in the text.

**L357 & Table 1: would it be clearer to reserve the term 'control run' for the definition in Eq 2 only and call the 'non-assimilative run' the 'free run' throughout the text?**

It would, I agree. I will change that.

**L391: What's the DA impact for depths below 250 m?**

As suggested above, I will remove this section. But the impact reduces quickly below 250 m, as can be seen in Fig. 4.

**Figure 7 does not include O2 or pH while Figure 8 does. What's the rationale of presenting different set of variables here?**

Fig. 5-7 were chosen to match the variables shown in Fig. 2, with extra variables just presented in Fig. 8 to limit the number of figures. I agree that it would be better to expand the range of information presented in this section, as suggested in detail by Referee #2, and so will present extra assessment accordingly.

**L522: Any comment on why O2 is not improved by BGC-Argo data? And why "in situ technologies such as gliders" can play a role?**

I think the likely reason that surface $O_2$ and chl-a are not really improved by BGC-Argo data, whereas surface $NO_3$ and pH are, is due to the relative importance of top-down versus bottom-up control for these variables, and the density of data required for the assimilation to have a major impact. In the case of $NO_3$, and DIC which helps control pH, concentrations typically increase with depth, and the supply of $NO_3$ and DIC from below the mixed layer is a major contribution to surface values. Therefore, changes at depth due to the assimilation will alter surface values through indirect processes. $O_2$ and chl-a typically decrease in concentration with depth, and dynamics within the mixed layer are much more important in setting surface values. For $O_2$, major drivers are temperature and ocean-atmosphere exchange. For chl-a a major driver is light availability. It seems that the BGC-Argo data is too sparse, even in ARGO_FULL, to have a widespread impact in these circumstances. More dedicated observations within the mixed layer may be likely to have more of an impact on surface values. For chl-a, this can be provided by ocean colour. For $O_2$, an obvious candidate is gliders. I will add a fuller description of these points in the revised text.

---

## Author Response (AR1)

**General comments:**
**This study tackles an important question regarding the potential benefit of assimilating BGC-Argo profiles in improving the global ocean biogeochemical reanalyses. The manuscript consists of 3 sections: 1) establish an OSSEs framework; 2) assess different strategies of updating BGC model state variables when assimilating ocean color data; 3) evaluate the benefit of assimilating different numbers of BGC-Argo profiles in addition to ocean color data over the assimilation of profiles or ocean color data alone. Overall, I think this work is well-conceived and will make an important contribution to the state-of-art ocean biogeochemical data assimilation combining the routinely available ocean color data and the emerging BGC-Argo observations.**

Thank you for your positive assessment and constructive comments. I will address each of these in turn below.

**I have two major comments. First, I feel that the second section of the manuscript on different DA strategies, at its present form, does not add much value to the story and the selection of best DA strategy involving a nitrogen balancing scheme before thorough tuning doesn't seem fair. Second, the first section on OSSEs requires a bit more analysis to prove its credibility. I'll provide more detailed explanations below. Aside from that, I have some minor comments, mostly technical, for the author to consider.**

Thank you for these comments. On reflection, I agree that the section on ocean colour assimilation strategies does not add much value to the core aims of the manuscript, and is not fully fledged. I have therefore removed this section, just presenting the results of the OC_3D_PHY run alongside the BGC-Argo assimilation. I will potentially develop the ocean colour assimilation work further in a future publication.

I also agree that presenting further analysis of the OSSE framework would help demonstrate its credibility. I have included the types of assessment requested, as detailed in turn below.

**Upon appropriately addressing these comments, I'll recommend publication of the manuscript in Biogeosciences.**

**1. I question on the value of including section 2 on comparing different update strategies when assimilating surface chl data for following reasons:**

**1) While I acknowledge the efforts and time needed for comparing 6 different DA strategies, I feel that the present comparison is not sufficient for fairly selecting the best DA strategy. I would argue that the more sophisticated nitrogen balancing scheme failing to outperform other strategies is largely because the parameter values used in the scheme are directly adopted from Hemmings et al. (2008) without proper tuning. These parameters reflect the BGC model's inherent relationships between chl and other model state variables. Since the model used here (MEDUSA) has quite different structure from that of Hemmings (HadoCC), a careful calibration of the parameters in the N balancing scheme is needed before its usage. That maybe contribute to a separate manuscript focusing on the benefit of multivariate BGC update over single-variable update.**

I agree with this assessment. Recalibrating the parameterisations of the Hemmings et al. (2008) nitrogen balancing scheme for specific use with MEDUSA would be a considerable amount of work, but necessary to achieve the best results. As suggested, I have therefore

removed this section from the current manuscript, and will potentially develop the work further as part of a separate paper. I now restrict discussion of different multivariate balancing options to the following in Section 5:

*"An alternative approach could be to use the nitrogen balancing scheme of Hemmings et al. (2008), which explicitly updates several model state variables to try and account for differing errors in phytoplankton growth and loss processes. This has been successfully used in previous ocean colour assimilation studies (Ford et al., 2012; Ford and Barciela, 2017; Ford, 2020) with the HadOCC model (Palmer and Totterdell, 2001). It was originally designed and tuned for use with HadOCC, so requires further development and tuning for use with the more complex MEDUSA, but an initial implementation for MEDUSA has been developed. Such a scheme offers more potential for controlling the wider biogeochemical state, especially if it could be expanded to alter parameter values as well as state variables."*

**2) At its present form, I didn't see strong connection between section two and three in the manuscript. To me, the most significant findings are from section three and this section stands out even if section two is completely removed. This is because the comparison in section two didn't suggest a clear winner and the ultimate decision of using the DA strategy of an intermediate complexity rather than the most sophisticated one (the N balancing scheme) for section three further reduce the value of including the entire comparison in section two.**

I agree that the most significant findings, in relation to the core aims of the study, are in the final section, and that this section deserves the most attention.

**3) If section two was removed, the author can have more space to elaborate and focus on the impacts of assimilating BGC-Argo profiles on different variables and suggesting directions for future work to improve. Currently, I feel the discussion on this part is relatively short compared to the emphasis it receives in the title, abstract and Introduction.**

I agree. I have therefore removed the ocean colour section, and used the extra space to further develop the assessment and discussion surrounding BGC-Argo assimilation, as detailed below and in response to Referee #2.

**2. I think section one on establishing the OSSEs framework is key to the credibility of assessment on assimilation impact. Presently the only analysis provided to show the credibility of OSSEs is a comparison of the errors between FREE and OBS and between FREE and NATURE for surface chl, NO3 and pCO2 in Figure 2. According to the criteria of designing rigorous ocean OSSE system detailed in Halliwell et al. (2014), I would request the author to comment and/or provide some information on following aspects:**

**1) Can the NATURE run reasonably capture the key features measured by the observing systems (in this case the surface chla, and the BGC profiles)? The author refers the performance of NATURE run to references given in Section 2 which is not very clear to me which one exactly has the same configuration and time period as the one here. A brief summary and/or some figures on the performance of the NATURE run will help.**

The references in Section 2 detail assessment of the model components used, but not of the specific NATURE run which is newly presented in this manuscript. I have clarified this in the text:

*"Validation of the general performance of the different system components can be found in the references given in Section 2, and validation of the nature run is presented in Section 4.1."*

I have also presented some figures and assessment in Section 4.1 of the revised manuscript demonstrating the performance of the NATURE run as requested. The new Fig. 2 and Fig. 4, and surrounding text, compare surface fields and zonal mean sections between observation-based products, NATURE, and FREE.

**2) Figure 2 only presents the surface comparisons. Since the assessment involves the vertical profiles, can the author also comment on whether the errors between FREE and NATURE are comparable to those between FREE and OBS in terms of the vertical distribution pattern of observable BGC variables.**

I have added a new Fig. 4, with surrounding assessment in Section 4.1, which shows zonal mean sections down to 2000 m for BGC variables for which there is an observation-based product to compare against. The errors between FREE and NATURE and FREE and OBS are broadly comparable, but often slightly smaller between FREE and NATURE. This is noted and discussed in the revised manuscript:

*"For regions and variables where the errors between FREE and NATURE were too low, the potential result may be to underestimate the impact of assimilating dense data, in this case ocean colour, and overestimate the impact of assimilating sparse data, in this case BGC-Argo (Halliwell et al., 2014). This should be borne in mind when drawing conclusions."*

**3) How about the error growth rate? One important criterion of credible OSSE evaluation is that the differences between the FREE and the NATURE ("truth") grow at the same rate as errors that develop between the state-of-the-art ocean models and the true ocean (Halliwell et al. 2014).**

I agree this is important to demonstrate, and I have added a new Fig. 5, with surrounding assessment in Section 4.1, demonstrating that this is the case.

**Halliwell, G. R., Srinivasan, A., Kourafalou, V., Yang, H., Willey, D., Le Hénaff, M., and Atlas, R.: Rigorous evaluation of a fraternal twin ocean OSSE system for the open Gulf of Mexico, J. Atmos. Ocean Tech., 31, 105–130, https://doi.org/10.1175/JTECH-D-13-00011.1, 2014.**

**Specific comments:**

**L21: '… half the planet's primary production.' Reference?**

I have added a reference to Field et al. (1998, http://doi.org/10.1126/science.281.5374.237).

**L75-77: Could you briefly add the outcome of assimilating these BGC-Argo observations in these two studies?**

I have added the following:

*"For instance, BGC-Argo observations of $O_2$ have been assimilated by Verdy and Mazloff (2017), who produced a five-year state estimate of the Southern Ocean using an adjoint method, and were able to capture over 60% of the variance in oxygen profiles at 200 m and 1000 m depth. Furthermore, Cossarini et al. (2019) assimilated BGC-Argo profiles of Chl-a into a model of the Mediterranean Sea, and found this was successful in adjusting the shape*

*of chlorophyll profiles, and that with the present number of BGC-Argo floats they could constrain phytoplankton dynamics in up to 10% of the Mediterranean Sea."*

**L87-88: 'Two groups perform … and the Met Office … presented here.' this information may be meaningful for the groups involved but doesn't seem informative for general readers. Do the two groups aim at different perspectives of the BGC assessment? What are they then?**

I have rephrased this section to be less focussed on the details of the project, and instead give a brief summary of the results of Germineaud et al. (2019):

*"Two groups performed OSSEs assessing biogeochemistry, Germineaud et al. (2019) and this study. Germineaud et al. (2019) presented a probabilistic evaluation at a single assimilation time step, finding that Chl-a from BGC-Argo floats added value at surface locations where ocean colour was unavailable, and at depth."*

**L116-118: Isn't that oxygen and dissolved inorganic N are also simulated? Or are they implicitly included in the 'coupled carbon cycle'? It's not clear to me what 'coupled' means here.**

This sentence was poorly phrased and incomplete. Oxygen and DIN are indeed also simulated, and the word "coupled" was unnecessary and confusing in this context. The revised text reads:

*"MEDUSA is of intermediate complexity, representing two phytoplankton and two zooplankton types, and the cycles of nitrogen, silicon, iron, carbon, and oxygen."*

**L154: This is acceptable, but could you comment if the physics of the NATURE run without DA is reliable to conduct the OSSEs?**

The NATURE run does capture key features of the physics. To demonstrate this, I have added an assessment of surface temperature to the new Fig. 2 and Fig. 3, with surrounding assessment in Section 4.1.

**L156: Just curious if there is any particular reason for using log10 instead of log-normal transformation.**

In practice, it should make no difference to the assimilation whether log-normal or $\log_{10}$ is used. The shape of the distribution is the same (the ratio of $\log(x)$ to $\log_{10}(x)$ is identical for all values of x), except that $\log_{10}$ gives a smaller variance. It is the shape of the distribution that matters for the assimilation, so as long as the same transformation is applied consistently to both model and observations, it should not matter whether log-normal or $\log_{10}$ is used. In the literature, some studies use $\log_{10}$ (e.g. Gregg, 2008, https://doi.org/10.1016/j.jmarsys.2006.02.015), while others use log-normal (e.g. Ciavatta et al., 2011, https://doi.org/10.1029/2011JC007219). The decision to use $\log_{10}$ here is a historical one, following Ford et al. (2012, https://doi.org/10.5194/os-8-751-2012), but as stated the choice should make no difference.

**L160: How large is the correlation length-scale? Water et al. (2015) is on physical DA. Same length scale used for the BGC assimilation here? I'm thinking that BGC fields are more dynamic and thus have a smaller correlation length-scale.**

The correlation length-scale is the same as in Waters et al. (2015, https://doi.org/10.1002/qj.2388), and varies with the Rossby radius, from a value of 25 km at low latitudes to 150 km at the Equator (see Fig. 3 of Waters et al., 2015,

https://doi.org/10.1002/qj.2388). For a 1/4° resolution ocean model, which is limited in its resolution of mesoscale features, using the same correlation length-scale for BGC as for physics is probably appropriate for an initial implementation. It is true though that the appropriate correlation length-scale(s) to use for assimilating BGC-Argo is an open question, and this should be addressed in future development of the assimilation.

I have clarified the length-scale in Section 2.2.2:

*"In the horizontal, a correlation length-scale based on the first baroclinic Rossby radius was used, varying from a value of 25 km at low latitudes to 150 km at the Equator, consistent with Waters et al. (2015)."*

I have also added the following to Section 5:

*"Related to this, the correlation length-scales used by the assimilation should be appropriately tuned for biogeochemical variables. In this study, a single horizontal correlation length-scale based on the first baroclinic Rossby radius was used, varying from a value of 25 km at low latitudes to 150 km at the Equator, following the physics implementation of Waters et al. (2015). This may help explain why the BGC-Argo assimilation demonstrated less widespread impact at high latitudes than near the Equator. A different correlation length-scale may be appropriate for biogeochemical variables. Furthermore, NEMOVAR has recently been developed to allow the use of multiple correlation length-scales (Mirouze et al., 2016), so both small- and large-scale corrections can be considered."*

**L165-170: Can the surface information help constrain the BGC fields below the mixed layer? '… below the mixed layer the vertical length-scale increases with the model's vertical grid resolution.' this is confusing to me.**

The length-scales are designed to limit the spreading of information across the base of the mixed layer. For an example see Fig. 4 of Waters et al. (2015, https://doi.org/10.1002/qj.2388) and the surrounding description in their Section 3.6. For surface observations the length-scale is set equal to the mixed layer depth, meaning that information from the surface observations is spread to the base of the mixed layer, but has limited impact on BGC fields below it. This is a deliberate decision based on the lack of correlation between water mass properties in and below the mixed layer. This is likely to be the case as much for BGC as for physics (see e.g. Fontana et al., 2013, https://doi.org/10.5194/os-9-37-2013). The vertical correlation length-scale is set to a minimum value at the mixed layer depth, and then increases with depth (see Fig. 4 of Waters et al., 2015, https://doi.org/10.1002/qj.2388). This increase is proportional to the increase in vertical model grid spacing that occurs with depth.

I have rephrased the description in the text to make this clearer. In Section 2.2.2:

*"The vertical correlation length-scale is dependent on the model's mixed layer depth, as determined from a one-day model forecast. At the surface, the vertical correlation length-scale is set to the depth of the mixed layer, so that information from surface observations is spread to the base of the mixed layer but not below it."*

In Section 2.2.3:

*"The vertical correlation length-scale was flow-dependent and varies with depth, as detailed by Waters et al. (2015). At the surface the vertical correlation length-scale was set to the depth of the mixed layer, decreasing to a minimum value at the base of the mixed layer. This minimised the spread of information across the pycnocline, due to the lack of correlation of water mass properties in and below the mixed layer (Waters et al., 2015; Fontana et al.,*

*2013). Below the mixed layer, the vertical correlation length-scale increased with depth, proportional to the increase in vertical model grid spacing that occurs with depth."*

**L172: 'The increments … from the two methods should be similar, though not identical.' Why? Isn't that the two methods have different treatments below the mixed layer?**

As I have removed the section comparing ocean colour assimilation strategies, I have also removed this sentence.

**L191: are these ratios fixed or time-dependent?**

Again, as I have removed the section comparing ocean colour assimilation strategies, I have also removed this section. But to answer the question, the ratios are time-dependent, based on the background ratios in each assimilation cycle.

**L216-217: this sentence should be reworded, something like: 'The approach taken to the assimilation of partial pressure of CO2 (pCO2) into HadOCC (While et al., 2012) is therefore adopted here with pH. In HadOCC, pCO2 is a function of temperature, …'**

I have reworded the sentence as suggested:

*"The approach taken to the assimilation of partial pressure of $CO_2$ ($pCO_2$) into HadOCC (While et al., 2012) was therefore adopted here with pH. In HadOCC, $pCO_2$ is a function of temperature, salinity, DIC, and alkalinity, and at constant temperature and salinity constant lines of $pCO_2$ are found in DIC/alkalinity space (see Fig. 1 of While et al. (2012))."*

**L261: Fujii et al. 2019 suggested the assimilative model to be configured either in reduced resolution or sufficiently different physical parameterizations.**

I have clarified this in the text:

*"The nature run is often run either at higher resolution than the assimilative model, or with significantly different parameterisations (Fujii et al., 2019)."*

**L272: 'year 5000', is it true or typo?**

This is true. Spinning up UKESM1 for CMIP6 was a massive endeavour, as documented by Yool et al. (2020, https://doi.org/10.1029/2019MS001933).

**L311: 30% is fine for estuarine and coastal waters, but would it be too large for chl-a profiles in open ocean?**

30% is a commonly used value in open ocean chlorophyll assimilation studies (e.g. Pradhan et al., 2020, https://doi.org/10.1029/2019JC015586), and especially for daily products I would expect this to be appropriate. For instance, Krasemann et al. (2017, https://esa-oceancolour-cci.org/sites/esa-oceancolour-cci.org/alfresco.php?file=6d534e45-fbfd-4cc5-8125-d84f0b3abea6&name=OC-CCI-PVIR-v3.20170303.pdf) found an RMSD of 0.31 for ocean colour matchups of $\log_{10}$(chl-a) against in situ observations. Maritorena et al. (2010, https://doi.org/10.1016/j.rse.2010.04.002, Fig. 10) estimated errors to be in excess of 30% across much of the ocean for daily products, though lower for monthly composites.

I have modified the text as follows:

*"A standard deviation of 30 % was agreed on within AtlantOS for Chl-a from ocean colour, a value commonly used in assimilation studies (Pradhan et al., 2020)."*

**L327: for these variables, are the error standard deviations fixed or monthly varying as well?**

They are fixed, I have made this clearer in the text:

*"Observation error standard deviations were set to a climatological constant equal to the average global observation error specified. These were fixed in time, and specified as …"*

**L357 & Table 1: would it be clearer to reserve the term 'control run' for the definition in Eq 2 only and call the 'non-assimilative run' the 'free run' throughout the text?**

It would, I agree. I have changed that throughout the text.

**L391: What's the DA impact for depths below 250 m?**

As suggested above, I have removed this section. But the impact reduces quickly below 250 m, as could be seen in the original Fig. 4.

**Figure 7 does not include O2 or pH while Figure 8 does. What's the rationale of presenting different set of variables here?**

Fig. 5-7 were chosen to match the variables shown in Fig. 2, with extra variables just presented in Fig. 8 to limit the number of figures. I agree that it would be better to expand the range of information presented in this section, as suggested in detail by Referee #2, and so I have presented extra assessment accordingly. The original Fig. 5-7 have now been amalgamated into a new Fig. 7, with $O_2$ and pH also added. Surrounding assessment is presented in Section 4.2.

**L522: Any comment on why O2 is not improved by BGC-Argo data? And why "in situ technologies such as gliders" can play a role?**

I have added the following to Section 5:

*"All assimilated variables were greatly improved below the mixed layer, but at the surface more limited improvements were seen for Chl-a and $O_2$, than for $NO_3$ and pH. This is likely due to the relative importance of top-down versus bottom-up control for these variables, and the density of data required for the assimilation to have a major impact. In the case of $NO_3$, and DIC which helps control pH, concentrations typically increase with depth, and the supply of $NO_3$ and DIC from below the mixed layer is a major contribution to surface values. Therefore, changes at depth due to the assimilation will alter surface values through indirect processes. $O_2$ and Chl-a typically decrease in concentration with depth, and dynamics within the mixed layer are much more important in setting surface values. For $O_2$, major drivers are temperature and ocean–atmosphere exchange. For Chl-a a major driver is light availability. It seems that the BGC-Argo data was too sparse, even in ARGO_FULL, to have a widespread impact in these circumstances. More dedicated observations within the mixed layer may be likely to have more of an impact on surface values. For Chl-a, this can be successfully provided by ocean colour, as the results of this study show. For $O_2$ and other variables, alternative in situ observing technologies such as gliders may be able to play a role (Telszewski et al., 2018)."*

Response to Referee #2

**The manuscript by Ford presents an OSSE experiment to investigate a number of assimilation strategies for ocean colour (OC) and biogeochemical Argo (BGC-Argo) observations using an already published DA method. The simulations are performed using a global model and sets of synthetic observations that resemble the current L3 chlorophyll OC and two potential arrays of BGC-Argo based on the current Argo network.**

Thank you for your review and constructive comments. I will answer these in turn below.

**The manuscript is well written and the performed modelling experiments allow novel and useful insights on the integration of BGC-Argo and OC data into global model assimilation. However, as presented, results seem rather superficial. The work would be the basis for a very valuable paper, but the results need to be explored further before I can recommend publication. My main concern is that results present a single month of simulation (i.e., the last month of a 2-year simulation: 1-year spinup and 1-year assimilation) and a single global statistics (e.g., Figs. 3, 4 and 10). Conclusion/discussion on assimilation strategies and impact of the observing systems are possibly misled by the limited results. Results of the whole year of the assimilation runs should be presented and the MEAred maps and profiles enriched with spatial and temporal statistics to provide quantitative insights on the impact of BGC-Argo in different seasons and regions of the global ocean. In particular, which are the areas and seasons that could benefit most from BGC-Argo assimilation and the integration of the two observing systems? I feel that the manuscript misses the objective to provide useful indications to design future observing system strategy, as it is proposed in the title.**

Thank you for your comments. I agree that more detailed assessment would be useful, and have greatly expanded the assessment presented in Section 4.2, as suggested.

**A second issue concerns the comparison between the PHY (phytoplankton update) and NIT (nitrogen balancing scheme) assimilation schemes. While the novelty of the OSSE experiment is related to the integration of OC and BGC Argo, the lack of the NIT implementation for BGC-Argo assimilation is a significant limitation of the manuscript that should be discussed. L425-427 are misleading. In fact, the choice of the PHY update scheme is explained at L229-230 (i.e., apparently, the NIT method has not been implemented for the joint BGC-Argo and OC assimilation). The joint OC and BGC-Argo assimilation strategy should be clearly described at L347-348. Then, it is not clear the objective of the first set of experiments (OC assimilation, which conclusions are mainly already published) if its results are not used for the second set of experiments. Even if the NIT method has not been implemented, the manuscript can be completed by a more unbiased discussion on pro and cons of the two methods and by presenting the work to be done and the the benefits to have the NIT method working with the BGC-Argo assimilation. In fact, some conclusions seem misleading. While the nitrogen balancing scheme is reported as the method with more potentiality (L406-407 in results and L510-L515 in discussion), results on BGC-Argo assimilation runs do not support this conclusion. I suggest to clarify better the proposed assimilation strategy and to detail better what would be needed to have the NIT method working with BGC-Argo assimilation. The conclusion that "only minimally altered for use with MEDUSA, so more specific tuning may help (L516-517)" seems misleading.**

Referee #1 also questioned the value of including the comparison of different ocean colour assimilation strategies in this manuscript. Based on the feedback of both referees, I have

removed this section of the paper. This should result in a more focussed manuscript and avoid the confusion my description of the different strategies seems to have caused. It has also allowed more space to further explore the results of the BGC-Argo experiments. I now restrict discussion of different multivariate balancing options to the following in Section 5:

*"An alternative approach could be to use the nitrogen balancing scheme of Hemmings et al. (2008), which explicitly updates several model state variables to try and account for differing errors in phytoplankton growth and loss processes. This has been successfully used in previous ocean colour assimilation studies (Ford et al., 2012; Ford and Barciela, 2017; Ford, 2020) with the HadOCC model (Palmer and Totterdell, 2001). It was originally designed and tuned for use with HadOCC, so requires further development and tuning for use with the more complex MEDUSA, but an initial implementation for MEDUSA has been developed. Such a scheme offers more potential for controlling the wider biogeochemical state, especially if it could be expanded to alter parameter values as well as state variables."*

**Minor points:**
**Line 173: why should the two methods provide similar increments? One is uniform with depth from surface to MLD depth, while the second method is not limited to the MLD and vertical increments are mediated by covariance.**

I have removed this sentence, as it was a source of confusion. It is true that in the 3D method vertical increments are mediated by covariance, which will be a source of differences. To clarify though, in the 3D method the vertical correlation length-scales are defined to allow surface information to be spread to the base of the mixed layer but not below it. I have rephrased the description of the assimilation method to make this clearer. In Section 2.2.2:

*"The vertical correlation length-scale is dependent on the model's mixed layer depth, as determined from a one-day model forecast. At the surface, the vertical correlation length-scale is set to the depth of the mixed layer, so that information from surface observations is spread to the base of the mixed layer but not below it."*

In Section 2.2.3:

*"The vertical correlation length-scale was flow-dependent and varies with depth, as detailed by Waters et al. (2015). At the surface the vertical correlation length-scale was set to the depth of the mixed layer, decreasing to a minimum value at the base of the mixed layer. This minimised the spread of information across the pycnocline, due to the lack of correlation of water mass properties in and below the mixed layer (Waters et al., 2015; Fontana et al., 2013). Below the mixed layer, the vertical correlation length-scale increased with depth, proportional to the increase in vertical model grid spacing that occurs with depth."*

**Line 335: can the author provide some more details on how the observation and background errors have been matched? and which is the value of inflation?**

The background error standard deviations estimated using the Canadian Quick method were output on the model grid, and the global mean value calculated. For each variable, the estimated standard deviations were multiplied by a constant so that the global mean value now matched the constant observation error standard deviations of 0.638 mmol m$^{-3}$ for NO$_3$, 2.767 mmol m$^{-3}$ for O$_2$, and 0.006 for pH. This meant that the global mean of each field matched the observation error standard deviations, while maintaining the spatial variation of the original estimates. I have added the following:

*"In order to give sufficient weight to the observations for the assimilation to be effective, the background error standard deviations were inflated. This was achieved by multiplying the*

*gridded field of background error standard deviations for each variable by a constant, so that the global mean background error standard deviation matched the observation error standard deviation used for that variable. This meant that on average, equal weight was given to the background and to the observations, but the ratio of background to observation error varied spatially based on the estimates from the CQ method."*

**Line 336: this sentence is not clear: the observation error is set from the real global BGC-float array, so what will change when the system is functioning with real BGC-floats? More generally, the discussion missed to tackle this topic: how much are the results affected by the selected observation and background errors? would it be a different impact of the two observing systems using different observation errors?**

The observation error will remain the same, but the background error will change. In part, this will be due to the different model parameterisations used in the OSSE framework compared with the standard model setup. Furthermore, the background error should reflect the error in the assimilative system, which will be dependent on the number and locations of BGC-Argo floats in the real ocean. It is likely that a different specification of the observation and background errors would give different quantitative results, but show a similar qualitative impact.

I have modified Section 3.4:

*"Once the system is fully functioning with real BGC-Argo data available, the background error estimates can be appropriately refined, based on the errors in the real-world assimilative model, and the actual distribution of BGC-Argo observations."*

I have added the following to Section 5:

*"The background error standard deviation estimates also need to be refined once real BGC-Argo observations are being assimilated, to reflect the background error in the assimilative system, which will depend on the actual distribution of BGC-Argo floats. The average ratio of background to observation error may also differ from that assumed in this study. It is likely that this would give different quantitative results, but the qualitative impact of the assimilation would remain similar."*

**Table 3 can be enriched to improve the identification of the runs at a glance. I would suggest to add 2 new columns for the assimilated and updated variables, and to split the note column in two new ones: background error (i.e., vertical propagations: 2D and 3D) and type of increment.**

Removing the comparison between ocean colour assimilation strategies means much of this information is now redundant, and I have simplified the table accordingly to aid identification of the runs.

**L354-355: explain how MAEosse and MAEcontrol are computed for pixels in the maps of Figures 5-9 and for points of the profiles in Figures 3, 4 and 10. Which are the distribution compared?**

For the maps, the MAE is calculated independently for each model grid cell by calculating the absolute difference between the model run and the nature run on each day of the given time period (the 31 days of December in this case), and then calculating the median of those 31 values. For the profiles, at each model depth level the absolute difference between the model run and the nature run on each day of the given time period is calculated, giving a set of values comprising of 31 days x 1442 longitudes x 1207 latitudes (with land points then excluded). The median of this set of values, weighted by the area of each grid cell, is then

calculated to give the global MAE value for that depth level. I have moved detail of metrics to be a new Section 3.6, and added the following:

*"Where $MAE_{red\_abs}$ or $MAE_{red\_\%}$ is presented as a spatial map, the MAE was calculated independently for each model grid cell. This was done by calculating the absolute difference between the model run and the nature run in that grid cell on each day of the given time period, and calculating the median of those values. Where $MAE_{red\_abs}$ or $MAE_{red\_\%}$ is presented as a profile, the MAE was calculated independently for each model depth level. At each depth, the absolute difference between the model run and the nature run on each day of the given time period was calculated for each grid cell in the region of interest. The median of this set of values was calculated, weighted by the area of each grid cell, to give the MAE value for that depth level."*

**L377: the absolute differences in Atlantic and Indian Oceans appear significant (i.e., 1 order magnitude). Can the author provide more details about which modifications of the FREE run (w.r.t. NATURE run) should have served to increase nutrient concentration? And which modifications compensate it (L382)? I agree that achieving a global appropriate level of error with a complex model (with uniform parameterization) can not be managed, however the author should provide some details on which modifications didn't work as expected. This can be helpful in understanding the effectiveness of the data assimilation in those areas.**

The nutrient differences in the Atlantic and Indian Oceans are of an order of magnitude, but with low absolute values, typically increasing from O(0.01) to O(0.1) mmol N m$^{-3}$. It is difficult to pinpoint the exact cause of any given change, as several parameters have been altered, and these will have complex interactions depending on the underlying concentrations of different variables. One potentially significant change though is that the nutrient uptake half-saturation concentration for phytoplankton was greatly increased for nitrogen, and decreased for iron. In areas which are nitrogen-limited, phytoplankton will therefore be less efficient at taking up nutrients. Furthermore, a decrease in zooplankton grazing half-saturation concentration means zooplankton become more efficient at grazing low phytoplankton populations. In NATURE, the Atlantic and Indian Oceans are the areas with the lowest DIN and phytoplankton concentrations. A first-order explanation may therefore be that the increased nitrogen uptake half-saturation concentration means phytoplankton take up less DIN, resulting in higher DIN concentrations. This then allows greater phytoplankton growth, as more DIN is available, though it is used less efficiently. This is then balanced by an increase in grazing, resulting in slightly elevated DIN and zooplankton concentrations, but largely unchanged phytoplankton concentrations. In other areas, which aren't so nitrogen-limited, the balance of processes is different, leading to different changes.

I have added the following to Section 4.1:

*"The modifications introduced in FREE served to increase nutrient concentrations in these regions, but also to suppress phytoplankton growth, resulting in little overall change in Chl-a. This is largely the result of increasing the nitrogen nutrient uptake half-saturation concentration for phytoplankton, and decreasing the zooplankton grazing half-saturation concentration."*

**L385: the absolute difference of NO3 and pCO2 between FREE and NATURE seems much lower than that between FREE and real observation. Can the author discuss which are the implications of the low difference for the OSSE assimilation results? I would argue that the effectiveness of the assimilation might be limited in some areas by the low differences between the synthetic observations generated from NATURE and the FREE run. Further, I would argue that MAEred might not be a good metric**

**because of the MAEcontrol at the denominator in the areas where NATURE and FREE are so close.**

From previous studies, the conclusion has been that an insufficient level of error would lead to "an overestimation of impact when sparse data are assimilated and an underestimation when dense (e.g., satellite) data are assimilated" (Halliwell et al., 2014, https://doi.org/10.1175/JTECH-D-13-00011.1). This suggests that this study may overestimate the impact of BGC-Argo observations in some regions. I mentioned this in the original manuscript, but agree it's a point which deserves further discussion. I have added the following to Section 4.1:

*"For regions and variables where the errors between FREE and NATURE were too low, the potential result may be to underestimate the impact of assimilating dense data, in this case ocean colour, and overestimate the impact of assimilating sparse data, in this case BGC-Argo (Halliwell et al., 2014). This should be borne in mind when drawing conclusions."*

I have added the following to Section 5:

*"Furthermore, for some variables and regions the error between the free-running model and the nature run was smaller than might be expected in real-world systems, potentially leading to an overestimate of the quantitative impact of BGC-Argo data (Halliwell et al., 2014)."*

I have also added some assessment of the absolute as well as percentage reduction in MAE, to avoid the issue of having MAE$_{control}$ in the denominator. This gives generally similar conclusions, but I agree is a useful additional way of looking at the results. I have described the metric in the new Section 3.6, added a new Fig. 8, and mentioned this metric throughout the assessment in Section 4.2.

**L395:  Since large areas of the global ocean are characterized by a DCM below 60m depth, the degradation of MAE below 60m would deserve a more detailed comment.**

Given that phytoplankton biomass is not degraded, I would speculate that depth variations in carbon-to-chlorophyll ratios are not being correctly characterised in these runs. However, as I have removed the comparison of ocean colour assimilation strategies, and the remaining OC_3D_PHY run does not show this degradation, I have also removed this discussion from the revised manuscript.

**L405: why should the similar behaviours of OC_2D and OC_3D demonstrate that the use of NEMOVAR to create 3D increments for the combined OC and BGC-Argo float assimilation is fit-for-purpose?**

This comment was simply intended to imply that because OC_3D gives similar behaviour to the proven strategy of OC_2D, assimilating ocean colour in this manner (which is a prerequisite for combining OC and BGC-Argo chl-a) should also give acceptable results. As I have removed the comparison between ocean colour assimilation strategies, I have also removed this comment from the revised manuscript.

**Figure 4:  I would suggest to reduce y-axis to 0-250m depth for the chlorophyll, phytoplankton and zooplankton plots to increase their readability. Are the high positive values below 250 in chla and phytoplankton plots due to the very low values of those variables below the euphotic zone?**

I have altered the figure as suggested. The high percentage values are indeed due to the low absolute values, and I have added comment on this in the text:

*"Beneath the euphotic zone, where Chl-a was near-zero, the assimilation had little impact. Positive values of MAE$_{red\_\%}$ were seen below 250 m, but values of MAE$_{red\_abs}$ (not shown) tended to zero below about 220 m."*

**L450-451 and L477-478: Can the degradation of Alkalinity be due to an improper working of the smallest combined change of DIC and Alk with pH assimilation? A more detailed analysis is expected to show the pro and cons of the method when pH is used instead of pCO2.**

Testing of the scheme shows the calculation is being performed correctly, as also indicated by the overall improvement in pH. The cause is likely to be that making the smallest combined change to DIC and alkalinity is not necessarily the approach that minimises errors in both DIC and alkalinity. In some circumstances it might be more appropriate to e.g. make a smaller or no change to alkalinity, and a larger change to DIC. Or even to make a change of the opposite sign to alkalinity, and an even larger change to DIC to compensate. Unfortunately, without concurrent observations of DIC or alkalinity, this information is not known at the time of assimilation. This is equally the case whether pCO$_2$ or pH is being assimilated. An assumption therefore needs to be made, and during the development of the original pCO$_2$ assimilation scheme it was decided that the safest assumption would be to make the smallest combined change in DIC and alkalinity – an assumption adopted for pH in this study. In light of these results it may be worth revisiting this assumption, but to do so effectively would involve a great deal of experimentation which is best left for a future study.

I have added the following to Section 5:

*"A novel method for assimilating pH was introduced in this study, following the method for assimilating pCO$_2$ developed by While et al. (2012). The method corrects pH, a diagnostic variable in the model, by making the smallest combined change to DIC and alkalinity required to reach the target pH value. This was successful in improving both pH and DIC, but alkalinity was often degraded. This highlights that making the smallest combined change to DIC and alkalinity does not guarantee that errors in both DIC and alkalinity are minimised. In some circumstances it might be more appropriate to make a smaller or no change to alkalinity, and a larger change to DIC. Or even to make a change of the opposite sign to alkalinity, and an even larger change to DIC to compensate. Unfortunately, without concurrent observations of DIC or alkalinity, this information is not known at the time of assimilation. This is equally the case whether pCO$_2$ or pH is being assimilated, and so it was decided that the safest assumption would be to make the smallest combined change to DIC and alkalinity. In light of these results it may be worth revisiting this assumption."*

**L454-L455: why are the negative values of MAEred in the Atlantic and Indian Oceans (Fig. 5) related to the compensating errors introduced in FREE? Since the FREE and NATURE differences are very low in those areas (Fig. 2), the impact of the assimilating synthetic observations (generated from NATURE) should be negligible. I wonder whether the MAEred is not a good metric because of the MAEcontrol at the denominator for those areas.**

Below is the original version of Fig. 5, showing the percentage reduction in MAE, and an alternative version showing the absolute reduction in MAE. It is true that the response in the Atlantic and Indian Oceans is minimal in absolute terms. I have added a new Fig. 8 showing this, and modified the surrounding discussion in Section 4.2 accordingly. I have also used both percentage and absolute reduction in MAE throughout this section.

[Figure]

**Figures 8 and 10 seem redundant and not necessary. For example, the MAEred over OC_3D_PHY of ARGO_FULL_OC (Fig. 8a and Fig 10a) provides the same information (except for the normalization of denominator) of the difference between MAEred over FREE of ARGO_FULL_OC and MAEred over FREE of OC_3D_PHY (Fig. 4a and Fig 5a and c). I suggest that the relative impact of adding BGC-Argo can be shown by a new table of the MAEred over FREE numeric values. The table can report values for selected regions and different seasons/months providing indications of which areas of the global ocean and periods of the year can benefit most by the BGC-Argo assimilation.**

As suggested, I have removed these figures, and added new Fig. 10-12 showing the impact of the BGC-Argo assimilation throughout the year in different regions. Please see the revised Section 4.2 for the detailed assessment.

**L505-508: which parameter settings between NATURE and FREE caused the degradation of the other variables? Can additional details be added?**

The interaction between different parameter changes is complex, and varies depending on the underlying concentrations of each of the variables. The biggest impact on zooplankton though is likely to have come from alterations to the grazing half-saturation concentration, which was changed from 0.8 mmol N m$^{-3}$ in NATURE to 0.36 mmol N m$^{-3}$ in FREE for microzooplankton, and from 0.3 mmol N m$^{-3}$ in NATURE to 0.135 mmol N m$^{-3}$ in FREE for mesozooplankton. Other significant changes to the ecosystem dynamics are likely to have come from changing the nutrient uptake half-saturation concentrations for phytoplankton. For nitrogen, this was changed from 0.5 mmol N m$^{-3}$ in NATURE to 2.13 mmol N m$^{-3}$ in FREE for non-diatoms, and from 0.75 mmol N m$^{-3}$ in NATURE to 3.195 mmol N m$^{-3}$ in FREE for diatoms. For iron, this was changed from 0.00033 mmol Fe m$^{-3}$ in NATURE to 0.00011 mmol Fe m$^{-3}$ in FREE for non-diatoms, and from 0.00067 mmol Fe m$^{-3}$ in NATURE to 0.00022 mmol Fe m$^{-3}$ in FREE for diatoms.

I have added the following to Section 4.1:

*"The modifications introduced in FREE served to increase nutrient concentrations in these regions, but also to suppress phytoplankton growth, resulting in little overall change in Chl-a. This is largely the result of increasing the nitrogen nutrient uptake half-saturation concentration for phytoplankton, and decreasing the zooplankton grazing half-saturation concentration."*

I have added the following to Section 5:

*"It seems likely that this degradation occurred due to the changed MEDUSA parameter settings between NATURE and FREE, meaning that the underlying processes were altered such that identical concentrations of phytoplankton now led to different concentrations of other variables. Relevant perturbations included changes to the grazing half-saturation concentration for zooplankton, and nutrient uptake half-saturation concentrations for phytoplankton."*

**L510-511 please explain: this seems not supported by results or references.**

I have added the following:

*"This will also be the case for assimilation schemes which use ensembles to generate cross-correlations between Chl-a and other model variables. These schemes are reliant on the model relationships between variables being correct, as it is these model relationships which the cross-correlations are based on. If the response of zooplankton to an increase in phytoplankton in the model ensemble differs from that in the real ocean, then the cross-correlations used in the assimilation will lead to a zooplankton response which follows the (incorrect) model rather than the real ocean, in exactly the same way as seen in this study."*

**L522: BGC-Argo assimilation has a small and positive impact on O2 as shown in Figure 10f (BGC-Argo assimilation). The degradation of O2 at surface seems due to OC assimilation (see figure 4f).**

Agreed. This was clumsy wording on my part, and I have rephrased this in the revised manuscript:

*"All assimilated variables were greatly improved below the mixed layer, but at the surface more limited improvements were seen for Chl-a and O$_2$, than for NO$_3$ and pH."*

**L531-L535: DA method improvements are important, but the paper does not really tackle this aspect; thus, those lines may fit better in the introduction and not as the last conclusion.**

I believe that the Discussion section is the best place for this discussion, as it details future work and recommendations arising from the results presented. I acknowledge that the original manuscript could have done this more effectively, and have expanded the discussion around assimilation improvements, relating it better to the results. I also take the point that something else, such as recommendations on observing system strategies, would fit better as the last conclusion, and have changed this around accordingly.

**Assimilating synthetic Biogeochemical-Argo and ocean colour observations into a global ocean model to inform observing system design**

David Ford[1]

[1]Met Office, FitzRoy Road, Exeter, EX1 3PB, UK

**Correspondence:** David Ford (david.ford@metoffice.gov.uk)

**Abstract.** A set of observing system simulation experiments  was performed. These assessed the impact on global ocean biogeochemical reanalyses of assimilating chlorophyll from remotely sensed ocean colour, and  in situ observations of chlorophyll, nitrate, oxygen, and pH from a proposed array of Biogeochemical-Argo (BGC-Argo) floats. Two  potential BGC-Argo array distributions were tested: one where biogeochemical sensors are placed on all current Argo floats, and one where biogeochemical sensors are placed on a quarter of current Argo floats.  Assimilating BGC-Argo data  greatly improved model results throughout the water column. This included surface partial pressure of carbon dioxide ($pCO_2$), which  is an important output of reanalyses. In terms of surface chlorophyll, assimilating ocean colour effectively constrained the model, with BGC-Argo providing no added benefit at the global scale. The vertical distribution of chlorophyll was improved by assimilating BGC-Argo data. Both BGC-Argo array distributions gave benefits, with greater improvements seen with more observations. From the point of view of ocean reanalysis, it is recommended to proceed with development of BGC-Argo as a priority. The proposed array of 1000 floats will lead to clear improvements in reanalyses, with a larger array likely to bring further benefits. The ocean colour satellite observing system should also be maintained, as ocean colour and BGC-Argo will provide complementary benefits.

*Copyright statement.* The works published in this journal are distributed under the Creative Commons Attribution 4.0 License. This licence does not affect the Crown copyright work, which is re-usable under the Open Government Licence (OGL). The Creative Commons Attribution 4.0 License and the OGL are interoperable and do not conflict with, reduce or limit each other.

© Crown copyright 2020

[revised manuscript text omitted]
). ~~AtlantOS had the aim of transforming various loosely-coordinated components into a "sustainable, efficient, and fit-for-purpose" Integrated Atlantic Ocean Observing System (IAOOS), consistent with the Framework for Ocean Observing (Lindstrom et al., 2012). One work package focussed on observing system design studies, using OSSEs to assess potential future improvements to existing and forthcoming components of the IAOOS.the Institute of Environmental Geosciences (IGE), and the Met Office. The IGE experiments have been published by Germineaud et al. (2019) , and the Met Office experiments are presented here~~ 
[revised manuscript text omitted]
. (2008), as has been routinely used with HadOCC (Ford et al., 2012). This uses a principle of conservation of mass to calculate increments to the six HadOCC state variables (phytoplankton, zooplankton, dissolved inorganic nitrogen (DIN), detritus, dissolved inorganic carbon (DIC), alkalinity) at all depths. The scheme was designed and parameterised for use with HadOCC, so is not immediately compatible with the more complex and differently parameterised MEDUSA. In this study it has been extended for use with MEDUSA by summing each of the phytoplankton and zooplankton functional types to obtain total phytoplankton and zooplankton, and using these as inputs to the nitrogen balancing scheme, while maintaining the parameter values of Hemmings et al. (2008). The scheme then calculates 3D increments to phytoplankton, zooplankton, DIN, detritus, DIC, and alkalinity, which are applied to the model, with phytoplankton split into diatoms and non-diatoms, and zooplankton into meso-zooplankton and micro-zooplankton, so that the background ratios are conserved. An increment is applied to silicate that is equal and opposite to the increment applied to diatom silicon biomass, to conserve silicon. Detrital carbon is updated to preserve the ratio between detrital nitrogen and carbonIn its original form, the scheme accepts 2D surface chl-a increments, and calculates one set of increments within the mixed layer using mixed layer-averaged values of background phytoplankton biomass, growth and loss rates, then further increments beneath the mixed layer by scaling the mixed layer increments to the local background phytoplankton biomass. In this study the scheme has been further extended to accept 3D chl-a increments, and calculate a corresponding set of multivariate increments for every depth level using the background values at that depth. This allows the scheme to be used with either 2D or 3D chl-a increments from NEMOVAR.~~, following the approach of Teruzzi et al. (2014) and Skákala et al. (2018, 2020).

**2.2.3 BGC-Argo**

For in situ profiles of biogeochemistry, as might be obtained from BGC-Argo, sets of 3D increments  were calculated for each assimilated variable, following the physics implementation of Waters et al. (2015). The method  was the same as for calculating 3D ocean colour increments, as described above. The vertical correlation length-scale was flow-dependent and varies with depth, as detailed by Waters et al. (2015). At the surface the vertical correlation length-scale was set to the depth of the mixed layer, decreasing to a minimum value at the base of the mixed layer. This minimised the spread of information across the pycnocline, due to the lack of correlation of water mass properties in and below the mixed layer (Waters et al., 2015; Fontana et al., 2013). Below the mixed layer, the vertical correlation length-scale increased with depth, proportional to the increase in vertical model grid spacing that occurs with depth.

In this study  Chl-*a*, $NO_3$, $O_2$, and pH  were assimilated, but the methodology is simple to extend to other model variables. As for ocean colour assimilation,  Chl-*a* is the sum of diatom and non-diatom  Chl-*a*, and a log-transformation  was performed prior to assimilation. As described above, assimilation of  Chl-*a* from ocean colour and in situ profiles can be combined. $NO_3$ and $O_2$ are state variables in MEDUSA, taking $NO_3$ to be equivalent to the MEDUSA DIN variable, while pH is a diagnostic variable calculated using version 2.0 of the mocsy carbonate package (Orr and Epitalon, 2015), which implements the SolveSAPHE algorithm (Munhoven, 2013) for solving the alkalinity-pH equation.

The  Chl-*a* increments were applied using the stoichiometric balancing method described for ocean colour above. The $NO_3$ increments  were directly applied to the MEDUSA DIN variable, and the $O_2$ increments to the $O_2$ variable. As pH is a diagnostic variable, the pH increments cannot be applied directly.  The approach taken to the assimilation of partial pressure of $CO_2$ ($pCO_2$) into HadOCC (While et al., 2012)  was therefore adopted here with pH. In HadOCC, $pCO_2$ is a function of temperature, salinity, DIC, and alkalinity, and at constant temperature and salinity constant lines of $pCO_2$ are found in DIC/alkalinity space (see Fig. 1 of While et al. (2012)). The scheme of While et al. (2012) assumes that temperature and salinity are error-free (and can be directly updated by physical data assimilation if not), and therefore updates DIC and alkalinity. As there is no unique combination of DIC and alkalinity that gives a specific $pCO_2$ value, the smallest combined change to DIC and alkalinity is made in order to reach the target  $pCO_2$ value. The same approach  was taken here with pH, which in MEDUSA is a function of temperature, salinity, nutrients, latitude, depth, DIC, and alkalinity. In DIC/alkalinity space, locally constant lines of pH are found when considering the range of present oceanic conditions (see Fig. 1a of Munhoven (2013)). The scheme developed here therefore updates DIC and alkalinity, assuming the other contributors to pH to be error-free, by making the smallest combined change which would give the target pH.

~~In the case where chl-a is assimilated using the nitrogen balancing scheme of Hemmings et al. (2008), and profiles of NO_3 and pH are also assimilated, this gives two different sets of increments to NO_3, DIC, and alkalinity. This combination is not used in this study, but in the current implementation precedence would be given to the increments derived from profiles of NO_3 and pH, as these are more directly related to the available observations, and just these increments applied in this situation. Ideally though, further balancing between the different variables would be applied, which can be considered as a future development.~~

[revised manuscript text omitted]

~~For the other seven variables plotted, which are either not directly updated by the assimilation or are only updated by the Hemmings et al. (2008) balancing scheme, results are mixed. OC_2D_CHL and OC_3D_CHL have near-zero $MAE_{red}$ values for all variables, demonstrating that just updating chl-a has very little impact on the wider model state. There are only small differences between OC_2D_PHY and OC_3D_PHY, and between OC_2D_NIT and OC_3D_NIT, suggesting that the use of NEMOVAR to create 3D increments, as required for combining assimilation of chl-a from ocean colour and BGC-Argo, is fit for purpose. Similarly, the extension of the Hemmings et al. (2008) balancing scheme to accept 3D chl-a increments as an input appears successful. The remainder of this sub-section will therefore focus on comparing OC_3D_PHY and OC_3D_NIT~~remained comparable throughout the year, with higher RMSE in austral summer in both cases. The RMSE variability was typically smaller between FREE and NATURE though, suggesting this to be a source of error not fully captured in FREE.

~~OC_3D_PHY results in a large degradation of surface zooplankton biomass and $NO_3$, with $MAE_{red}$ values of -358 % and -92 % respectively. This is reduced in OC_3D_NIT to -106 % and -15 % respectively. In both runs, for zooplankton biomass $MAE_{red}$ increases towards zero with depth. For $NO_3$, more complex variation in $MAE_{red}$ is seen with depth, likely reflecting regional variations in the impact of the two assimilation strategies around the nutricline in particular. Both runs improve detrital nitrogen, with $MAE_{red}$ values at the surface of 35 % for OC_3D_PHY and 23 % for OC_3D_NIT. $MAE_{red}$ decreases more quickly with depth in OC_3D_PHY, but remains positive in both cases. $O_2$ is degraded in both runs, with a larger surface degradation in OC_3D_NIT. With the carbon cycle, DIC, alkalinity, and pH are all degraded in OC_3D_PHY. In OC_3D_NIT, DIC is further degraded near the surface, but alkalinity is now improved, with positive $MAE_{red}$ through the water column. The result on pH, a diagnostic which is a function of DIC and alkalinity, is that OC_3D_PHY and OC_3D_NIT have a near-identical degradation in $MAE_{red}$ of around -157 % at the surface, but OC_3D_NIT gives better results with depth.~~

 ~~phytoplankton biomass, and does not degrade any other variable. It is commonly agreed though that it is highly desirable to try and use ocean colour data to improve the wider model state (Gehlen et al., 2015; Fennel et al., 2019), and this clearly cannot be achieved by solely updating chl-a. Present and future reanalyses do and will make multivariate updates, which are difficult to validate due to the sparsity of in situ observations, and this should be accounted for when considering the potential impact of~~ While it is not ideal that the Chl-*a* errors differ in the Tropical Atlantic and Indian Oceans, and errors between FREE and NATURE were too low for some variables, achieving globally appropriate levels of error with a complex biogeochemical model with globally uniform parameterisations could not be managed within the resources of the project. Furthermore, the similarity of Chl-*a* between NATURE and FREE in some regions was due to the introduction of compensating errors in FREE, rather than a lack of model error. This itself is a common feature of reanalyses, which can result in data assimilation increasing overall error by correctly reducing one of a set of compensating errors, as demonstrated by Ford and Barciela (2017). For regions and variables where the errors between FREE and NATURE were too low, the potential result may be to underestimate the impact of assimilating dense data, in this case ocean colour, and overestimate the impact of assimilating sparse data, in this case BGC-Argo ~~on such reanalyses. The most commonly used method is to update the phytoplankton biomass to preserve stoichiometry (Teruzzi et al., 2014; Skákala et al., 2018). Since using the Hemmings et al. (2008) balancing scheme did not show a clear overall improvement in these tests, it was therefore decided to use OC_3D_PHY as the basis for OSSEs introducing the assimilation of BGC-Argo data, and to use this method when assimilating profiles of chl-a~~(
[revised manuscript text omitted]
, generally gave an improvement over just updating phytoplankton biomass, but resulted in further degradation of some variables. This has been successfully used in previous ocean colour assimilation studies (Ford et al., 2012; Ford and Barciela, 2017; Ford, 2020) with the HadOCC model (Palmer and Totterdell, 2001). It was originally designed and tuned for use with HadOCC, so requires further development and tuning for use with the more complex MEDUSA, but an initial implementation for MEDUSA has been developed. Such a scheme offers more potential for controlling the wider biogeochemical state, especially if it could be expanded to alter parameter values as well as state variables. Furthermore, it was designed for use with the simpler HadOCC model (Palmer and Totterdell, 2001), with which it has been successfully demonstrated (Hemmings et al., 2008; Ford et al., 2012), and only minimally altered for use with MEDUSA, so more specific tuning may help.

[revised manuscript text omitted]

---

## Author Response (AR2)

**The author has done a good job of responding to all the reviewers' comments (including mine), and the manuscript is greatly improved as a result. It reads very well, and includes useful additions from the first version. I have only some minor technical corrections and suggestions, and my recommendation is to accept it.**

Thank you very much for your reviews and constructive comments.

**Minor comments.**
**Line 138: a model spatial correlation is introduced without a clear link with the previous sentences explaining the DA method. How is the model spatial correlation used by the DA method?**

I have provided an additional reference and expanded the description to:

*"A diffusion operator is used to efficiently model spatial correlations (Mirouze and Weaver, 2010; Mirouze et al., 2016). Ten iterations of the diffusion operator are applied, simulating the matrix multiplication of an autoregressive correlation matrix, which provides a good approximation to a Gaussian correlation function (Waters et al., 2015)."*

**Line 143 which unbalance component is used?**

I have added the sentence:

*"The unbalanced component considers the uncorrelated component of each variable using univariate error covariances, while the balanced component considers correlations between variables."*

**Line 211 and Line 214: OSSEs investigate the potential impacts of future observing systems or new observations of current observing networks. Thus, I wonder whether the reference to "current observing network" (line 211) and to "the assimilative run, which assimilates synthetic observations representing current observing networks" (line 214) can be confusing. I would suggest to refer to future observing networks, which, in this specific study, has been designed starting from a 1-year of Argo trajectories.**

This is true, and so I agree the way it's worded could be confusing. Separation does need to be made though between future observing networks, and existing observations of current observing networks which are already routinely assimilated – in this case ocean colour. I have therefore rephrased these two lines respectively as:

*"Synthetic observations representing both existing routine observations and future observing networks, which are sampled from the nature run with appropriate errors prescribed."*

*"An assimilative run, which assimilates synthetic observations representing existing routine observations into the alternative model simulation."*

**Lines 383-384: this sentence is not clear: why should it matter when the FREE-NATURE difference is compared to the error between observation and FREE? Shouldn't the difference between FREE and NATURE already include all possible (and reasonable) sources of uncertainty in order to be used for OSSEs?**

The difference between FREE and NATURE includes all reasonable sources of model-based uncertainty, and the synthetic observations sampled from NATURE include all reasonable sources of observation-based uncertainty. But in this figure NATURE is plotted directly, which is the "truth" and so does not itself included the observation-based uncertainty, while the real-world climatologies plotted here do include observation-based uncertainty. I have rephrased this sentence to:

*"It should be remembered though that the observation-based products used here have large uncertainties themselves, while NATURE is the "truth" and so error-free."*

**Lines 385-386: this explanation is not entirely convincing. The differences between FREE and OBS (Fig.3i) are up to 0.1 (which would be a rather high acidification signal in just 5 years) while the rather low difference between FREE and NATURE (Fig. 3j) seems possibly related to the initialisations of DIC and alkalinity which are maybe too close.**

I have removed this sentence.

**Line 390: annual zonal section plots of the absolute differences can be added to Fig. 4 (as done in Figure 3). This would greatly facilitate the reading of this section.**

I have added a new Fig. 5 showing these.

**Line 355: it seems to me that the choice to show December 2009 results is because the largest DA impacts are at the end of simulation (as it is explained latter). If it is like that, it should be mentioned at this point.**

I have rephrased this sentence to:

*"These are plotted for the final month of the simulations, December 2009, which is when the largest cumulative impact from the data assimilation will be seen."*

**Line 447: I wonder whether the measurement and representation errors added to the synthetic observations in areas where FREE and NATURE are too close force the assimilation of those observations to move the assimilative runs far from the NATURE. This would explain the worsening of assimilative runs in areas with very low FREE-NATURE differences.**

This is possible, yes. I have expanded this sentence to read:

*"These correspond to areas where NATURE and FREE were almost identical, as seen in Fig. 3d, and so the observation errors were larger than the model error."*

**Line 459-462: looking at Figures 7s,t and 7x,y, it seems to me that that MAE_red% of pCO2 is worse than that of pH (i.e., red coloured areas in x and y are larger than in s and t). It is not straightforward that pCO2 improves when pH is assimilated, as it is also discussed latter (lines 575-584). It should be mentioned at this point before it is discussed in the final section.**

I have rephrased this sentence to:

*"Improvements to DIC and alkalinity when assimilating pH should also help improve $pCO_2$, which was not directly assimilated, though the details of the impact differs between the two variables."*

**Figures 10, 11 and 12. Could the author reverse the colorscale of these Hovmoeller diagrams? Since the blue colour means reduction of the MAE in figures 7-9, it would increase the readability to have the same also in Fig.10, 11 and 12.**
**Further, all negative values have a single blue tone. This does not help to understand how relevant negative values are. It would be better to expand negative scale.**

I have reversed the colour scale, and modified it to run from -100 to 100%.

**Lines 471-473: how significant is this small degradation? The choice of the color scale (i.e., only one blu tone) of Figure 10 seems misleading. Curves of OC, ARGO_1/4_OC and ARGO_FULL_OC of figure 6a are almost indistinguishable.**

It is small as stated, this is better shown by the revised colour scale.

**Lines 488-492: does the author have any explanation for the different impact on MAE reduction in the different oceans? Is there any evidence that can be discussed about the effectiveness of the BGC-Argo network and its assimilation in the different oceanic areas?**

This is likely related to the correlation length scales, as discussed in the final section:

*"Related to this, the correlation length-scales used by the assimilation should be appropriately tuned for biogeochemical variables. In this study, a single horizontal correlation length-scale based on the first baroclinic Rossby radius was used, varying from a value of 25 km at low latitudes to 150 km at the Equator, following the physics implementation of Waters et al. (2015). This may help explain why the BGC-Argo assimilation demonstrated less widespread impact at high latitudes than near the Equator. A different correlation length-scale may be appropriate for biogeochemical variables. Furthermore, NEMOVAR has recently been developed to allow the use of multiple correlation length-scales (Mirouze et al., 2016), so both small- and large-scale corrections can be considered."*

I've also added the following sentence after line 492:

*"Increments also spread further in tropical regions due to longer correlation length-scales and faster propagation time-scales."*

**Line 684-690: this paragraph is perhaps the only part that rises me some concerns. It is not clear to me what top-down or bottom-up control means in this context. In ecology, these terms are usually referred to the trophic food web interactions (e.g. phytoplankton dynamics is top-down controlled by zooplankton grazing or bottom-up controlled by nutrient supplied).**
**Instead, these terms are used here to explain some vertical relationships between layers: deep layer and mixed layer. Could the author explain these relationships better?**

I agree that "top-down" and "bottom-up" were a confusing choice of terms. I have rephrased this passage to:

*"This is likely due to the relative importance of different physical processes affecting these variables, and the density of data required for the assimilation to have a major impact. In the case of $NO_3$, and DIC which helps control pH, concentrations typically increase with depth, and the supply of $NO_3$ and DIC from below the mixed layer is a major contribution to surface values. Therefore, changes at depth due to the assimilation will alter surface values through indirect processes. $O_2$ and Chl-a typically decrease in concentration with depth, and dynamics within the mixed layer are much more important in setting surface values. For $O_2$,*

*major drivers are temperature and ocean–atmosphere exchange. For Chl-a a major driver is light availability."*

**Further, it seems to me that the greatest improvement of BGC-Argo assimilation over OC occurs at the deep chlorophyll maximum and nitracline depths, which is a pretty nice result to discuss.**

I have added the following text:

*"The greatest improvement was in terms of Chl-a around the nitracline, at depths where deep chlorophyll maxima are likely to be found. This should help quantify a major contributor to global primary production, which cannot be observed from space."*

**Assimilating synthetic Biogeochemical-Argo and ocean colour observations into a global ocean model to inform observing system design**

David Ford[1]

[1]Met Office, FitzRoy Road, Exeter, EX1 3PB, UK

**Correspondence:** David Ford (david.ford@metoffice.gov.uk)

**Abstract.** A set of observing system simulation experiments was performed.  This assessed the impact on global ocean biogeochemical reanalyses of assimilating chlorophyll from remotely sensed ocean colour, and in situ observations of chlorophyll, nitrate, oxygen, and pH from a proposed array of Biogeochemical-Argo (BGC-Argo) floats. Two potential BGC-Argo array distributions were tested: one where biogeochemical sensors are placed on all current Argo floats, and one where biogeochemical sensors are placed on a quarter of current Argo floats. Assimilating BGC-Argo data greatly improved model results throughout the water column. This included surface partial pressure of carbon dioxide ($pCO_2$), which is an important output of reanalyses. In terms of surface chlorophyll, assimilating ocean colour effectively constrained the model, with BGC-Argo providing no added benefit at the global scale. The vertical distribution of chlorophyll was improved by assimilating BGC-Argo data. Both BGC-Argo array distributions gave benefits, with greater improvements seen with more observations. From the point of view of ocean reanalysis, it is recommended to proceed with development of BGC-Argo as a priority. The proposed array of 1000 floats will lead to clear improvements in reanalyses, with a larger array likely to bring further benefits. The ocean colour satellite observing system should also be maintained, as ocean colour and BGC-Argo will provide complementary benefits.

*Copyright statement.* The works published in this journal are distributed under the Creative Commons Attribution 4.0 License. This licence does not affect the Crown copyright work, which is re-usable under the Open Government Licence (OGL). The Creative Commons Attribution 4.0 License and the OGL are interoperable and do not conflict with, reduce or limit each other.

© Crown copyright 2020

[revised manuscript text omitted]